# Smart facemask for wireless CO$_2$ monitoring

P. Escobedo [1,2], M. D. Fernández-Ramos [2,3], N. López-Ruiz [1,2], O. Moyano-Rodríguez [3],
A. Martínez-Olmos [1,2], I. M. Pérez de Vargas-Sansalvador[2,3], M. A. Carvajal [1,2,4],
L. F. Capitán-Vallvey [2,3] & A. J. Palma [1,2,4 ✉]

The use of facemasks by the general population is recommended worldwide to prevent the spread of SARS-CoV-2. Despite the evidence in favour of facemasks to reduce community transmission, there is also agreement on the potential adverse effects of their prolonged usage, mainly caused by CO$_2$ rebreathing. Herein we report the development of a sensing platform for gaseous CO$_2$ real-time determination inside FFP2 facemasks. The system consists of an opto-chemical sensor combined with a flexible, battery-less, near-field-enabled tag with resolution and limit of detection of 103 and 140 ppm respectively, and sensor lifetime of 8 h, which is comparable with recommended FFP2 facemask usage times. We include a custom smartphone application for wireless powering, data processing, alert management, results displaying and sharing. Through performance tests during daily activity and exercise monitoring, we demonstrate its utility for non-invasive, wearable health assessment and its potential applicability for preclinical research and diagnostics.

[1] ECsens, CITIC-UGR, Department of Electronics and Computer Technology, University of Granada, Granada, Spain. [2] Unit of Excellence in Chemistry Applied to Biomedicine and the Environment of the University of Granada, Granada, Spain. [3] ECsens, Department of Analytical Chemistry, University of Granada, 18071 Granada, Spain. [4] Sport and Health University Research Institute (iMUDS), University of Granada, 18071 Granada, Spain. ✉email: ajpalma@ugr.es

In an effort to mitigate the rapid worldwide spread of severe acute respiratory syndrome coronavirus 2 (SARS-CoV-2), the virus that causes the COVID-19 disease, facemask wearing has been widely imposed by the vast majority of countries[1]. The current recommendation by the World Health Organization (WHO)[2] only to some particular scenarios (e.g. crowded or closed places) was swiftly extended to almost any situation where not cohabitant people are involved in shared common spaces. Not until very recently, has a report been published where a simple mathematical model is proposed to demonstrate the efficiency of the limitation of the virus transmission under the condition of low viral load[3]. This issue has raised great controversy regarding the actual effectiveness of facemasks on disease control against their possible adverse effects[4–6]. It is not the aim of this report to directly tackle this controversy, but to design and evaluate a battery-free and wearable solution to determine carbon dioxide ($CO_2$) concentration into the dead space volume (DSV) of a filtering facepiece FFP2 facemask. However, it is worth noting that, according to very recent studies, SARS-CoV-2 is evolving towards more efficient aerosol generation and loose-fitting masks could provide only modest source control[7]. In physiological terms, DSV can be defined as the volume of ventilated air in which there is no significant gas exchange of oxygen ($O_2$) and $CO_2$[8]. While standard facemasks simply act as air filters for the nasal and/or mouth passage, the integration of sensors to control parameters of interest can be an added value to improve their usage and effectiveness.

Some works have reported negligible physiological effects of the increased $CO_2$ concentration into the DSV[9–12], whereas others have pointed out the opposite conclusion with a statistically significant drop in blood oxygen saturation accompanied by heart and respiratory rate increases, and user discomfort[13–17]. Nonetheless, an agreement has been reached in two related facts: the DSV extension[18,19] and an increased breathing resistance when using facemask[9,10]. Besides the well-established clinical technique of capnography for real-time monitoring of the concentration or partial pressure of $CO_2$ in the respiratory gases, some more portable hardware has recently been reported to measure this gas inside the mask DSV. Both home-made solutions[20] and more or less bulky and complex standard instruments have been proposed[21,22]. Although very reliable and high-resolution data can be gathered with these systems, they are limited to laboratory experiments because of their size and power requirements, thus not providing an actual wearable measurement system.

To address the shortcomings of the current technology in terms of better wearability and conformability, thus providing results closer to real conditions, we have developed a platform for gaseous $CO_2$ real-time determination. Although this non-invasive platform can be used in many other situations, such as indoor air quality assessment, thanks to its compact size, it has been located in the inner face of an FFP2 mask. The proposed sensing system is based on a stable opto-chemical sensor with a limit of detection of 140 ppm of $CO_2$ concentration and resolution of 103 ppm for measuring this gas concentration in the DSV of a facemask. Response and recovery times are below 1 s in this $CO_2$ concentration range, while sensor lifetime is 8 h, comparable with FFP2 facemask recommended usage time. Both the sensor and the conditioning circuitry have been fabricated onto a light and flexible polymeric substrate conforming to what is usually named as sensing tag with final dimensions of $60 \times 45$ mm$^2$. This tag is powered by the near-field communication link (NFC) of a smartphone with a custom-developed software application for data processing and result displaying and sharing.

In this report, we have first characterized the sensor response and the complete analytical specifications of the measurement system. The studies have included the compensation of the effects of temperature; the independence with ambient humidity; and the assessment of the system functionality under ambient light with different intensities (from ~200 lx up to 67 klx). Next, we have successfully compared our sensing tag with reference instruments in three different experiments: short-term and long-term performances in a static situation, and a graded cycling exercise test. In all cases, results are consistent with previous studies regarding measured $CO_2$ concentration. Finally, similar wearable systems are compared with our solution and some discussion about system performance and test results are carried out. This study demonstrates the application of an optical sensor with printed NFC-based sensing tags for non-invasive gas determination by means of wearable solutions with a high potential for clinical translation.

## Results

**General architecture of the system.** General design criteria are based on those of IoT devices: sensing, connectivity, integration, compactness, scalability, low-energy consumption, and affordable cost. The general concept of the developed system is shown in Fig. 1 and Supplementary Movie. The proposed system consists of an NFC tag printed on a transparent flexible substrate and located on the inner face of an FFP2-type facemask. The system allows $CO_2$ readout by means of a deposited chemical sensor with the fluorescent response, whose real-time results are shown in an NFC-enabled smartphone application when it is approached the tag (Fig. 1). The proposed opto-chemical $CO_2$ sensor is based on the combination of a stable inorganic phosphor whose luminescence is modulated by an acid–base indicator through an inner filter mechanism responding to the gaseous $CO_2$ acidity. This sensing cocktail has been optimized to reduce the interfacial tension and facilitate the permeation of $CO_2$ gas. The sensing NFC-based system is fully passive, meaning that no battery is needed to power it. Instead, the power supply is obtained through energy harvesting from the near electromagnetic field induced by an external NFC reader. In this case, the reader can be any NFC-enabled smartphone in combination with the custom-developed app. In this way, a measurement of the $CO_2$ gas concentration is available virtually immediately (i.e. in real-time) upon the approach of the smartphone, although the monitoring is not continuous since it needs the powering from the smartphone acting as the NFC reader. In return, the system is battery-free, which has been possible by designing the electronics using ultra-low power consumption components.

The aim of this system is the measurement of gaseous $CO_2$ concentration into the DSV of an FFP2 (N95) facemask. In brief, the process is carried out by biasing an ultraviolet (UV) light-emitting diode (LED) placed on an NFC tag, attached to the inner face of the mask, with the electromagnetic field provided by the smartphone NFC link. The energy harvested by the tag antenna is conditioned by the NFC chip, then being able to power the rest of the electronic components. When the UV LED is powered on, the $CO_2$ sensing membrane is excited and the luminescence intensity is measured to determine the $CO_2$ concentration through the red coordinate ($R$) of the RGB colour space. The luminescence intensity coming from the sensor is registered using a digital colour detector, and the collected data is sent to a microcontroller unit (MCU). This information is processed to obtain the value of the $R$ coordinate, computing the $CO_2$ concentration from the sensor calibration curve. Finally, results are displayed and further processing is carried out in the software application running in the smartphone. This general platform for luminescent sensing membranes can be adapted for other gases determination (i.e. oxygen), provided that a suitable opto-chemical sensor is deposited on the tag.

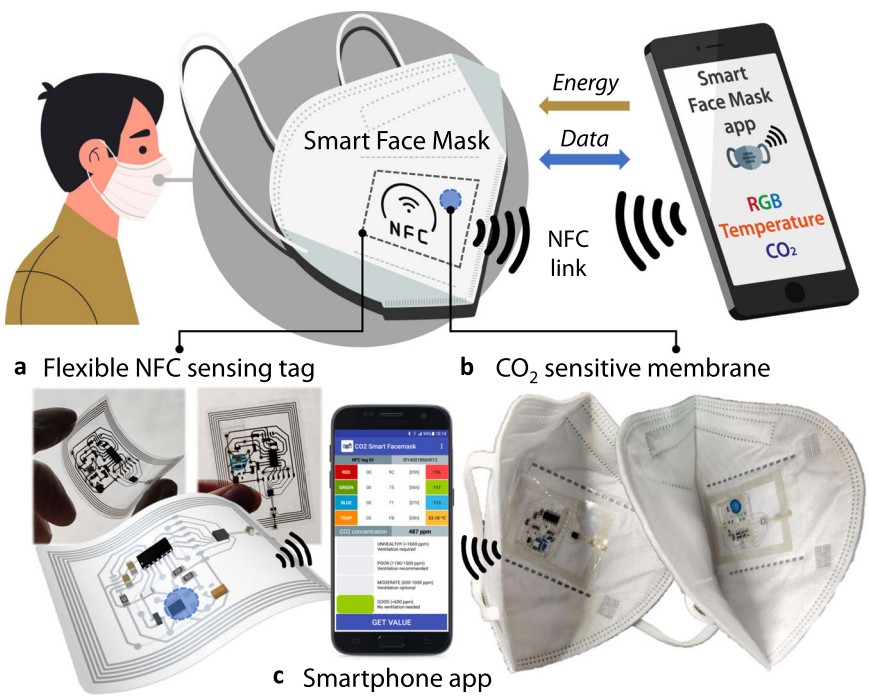

**Fig. 1 Overview of the NFC-based smart facemask for wireless $CO_2$ real-time determination. a** Flexible NFC sensing tag. **b** $CO_2$ sensitive membrane deposited on the flexible tag, which is attached to the inner layer of a standard FFP2 facemask. **c** Powering and bidirectional communication are achieved by means of custom-developed smartphone applications.

**$CO_2$ sensing scheme**. Existing methods for $CO_2$ sensing can be classified into four techniques: infrared absorptiometry, catalytic bead sensors, electrochemical devices, and optical sensors. Regarding infrared sensors, they are subject to more strong interference as well as an increase in the cost and the size of the final prototype due to the characteristics of available sensors. The catalytic bead sensors are based on flameless combustion, and the electrochemical sensors, although effective and reliable, can be cross-sensitive to other gases. Due to the requirement of low-power consumption of the proposed application, these techniques were discarded. Optical sensors (see a brief revision in Supplementary Table 1), based on the luminescence measurement of a very stable inorganic phosphorus modulated by an acid–base indicator depending on the $CO_2$ concentration, have been widely studied showing advantages that are really attractive for the developed system: possibility of miniaturization, electrical isolation and less sensitivity to noise.

The developed $CO_2$ sensor relies on the combination of an inorganic phosphor whose luminescence is modulated by an acid–base indicator, as graphically depicted in Fig. 2a. More details of this sensing approach are given in Supplementary Section 1. Lanthanum oxysulfide ($La_2O_2S$) is a wide-gap semiconductor material (~4.6 eV) suitable for doping trivalent rare-earth ions. The X-ray phosphor lanthanum oxysulfide:europium ($La_2O_2S$:Eu) was selected as luminophore, whose luminescent decay curve for $^5D_0 \rightarrow {}^7F_2$ transition (625 nm) of $Eu^{3+}$-doped $La_2O_2S$ spherical nanocrystals shows a long decay time of 1.2 ms[23] along with high chemical and photochemical inertness. As pH indicator, α-naphtholphthalein was selected, which shows adequate $pK_a$ value for gaseous $CO_2$ acidity and absorption spectra to act as an acceptor of $La_2O_2S$:Eu luminescence. In the presence of a hydrophobic quaternary ammonium hydroxide TMAOH, the pH indicator is in its basic form stabilized as a hydrated ion pair. As illustrated in Fig. 2b, the $La_2O_2S$:Eu phosphor shows a broad absorption band with a maximum peak at 325 nm, resulting from the charge-transfer transitions between oxide and sulfide ions and $Eu^{3+}$. Figure 2b also shows the emission spectra of the selected UV LED located at

367 nm, thus overlapping the absorption band of the $La_2O_2S$:Eu phosphor. The most intense emission, bright orange-red light, with a peak centred at 625 nm, corresponds to the forced electric dipole $^5D_0 \rightarrow {}^7F_2$ transition of $Eu^{3+}$ ions[24]. The deprotonated form of α-naphtholphthalein in HPMC is blue colour and its absorption spectrum (maximum 638 nm) strongly overlaps the luminescent emission spectrum of $La_2O_2S$:Eu at 625 nm, as depicted in Fig. 2b. Therefore, the working principle is based on an inner filter effect. In the presence of gaseous $CO_2$, the indicator is protonated, reducing the attenuation of the phosphor by the basic form of indicator, thus increasing the luminescence. This assumes a secondary inner filter effect mechanism that was confirmed by studying the luminescence lifetime decay variation with concentrations of $CO_2$ gas from 0% to 100%, observing a decay time value around 0.33 ms in the whole range. Decay times fitting was made via an exponential decay model.

$CO_2$ sensing membranes were deposited on 125 μm-thick polyethylene terephthalate (PET) flexible substrate of the NFC tag by spin coating. Prior to the fabrication, an optimization process of the optical sensor composition was conducted as described in Supplementary Section 1. To analytically characterize the fabricated membranes, the equation suggested by Nakamura and Amao[25] was used to theoretically verify the relationship between the luminescence intensity and the $CO_2$ concentration, as shown in Fig. 2c.

$$\frac{I_{100} - I_0}{I - I_0} = A \frac{1}{[CO_2]} + B \qquad (1)$$

where $I_0$ and $I_{100}$ are the intensities at 0% and 100% $CO_2$, respectively, and $A$, $B$ are proportionality constants. A good fit in the whole range, derived from three-fold replicates at room temperature, 25 °C, can be observed. The principal analytical performance characteristics of the luminescent $CO_2$ gas sensor are summarized in Supplementary Table 3. Figure 2d shows the dynamic response of the sensing membranes when exposed to alternative atmospheres of pure $CO_2$ and pure $N_2$. A response

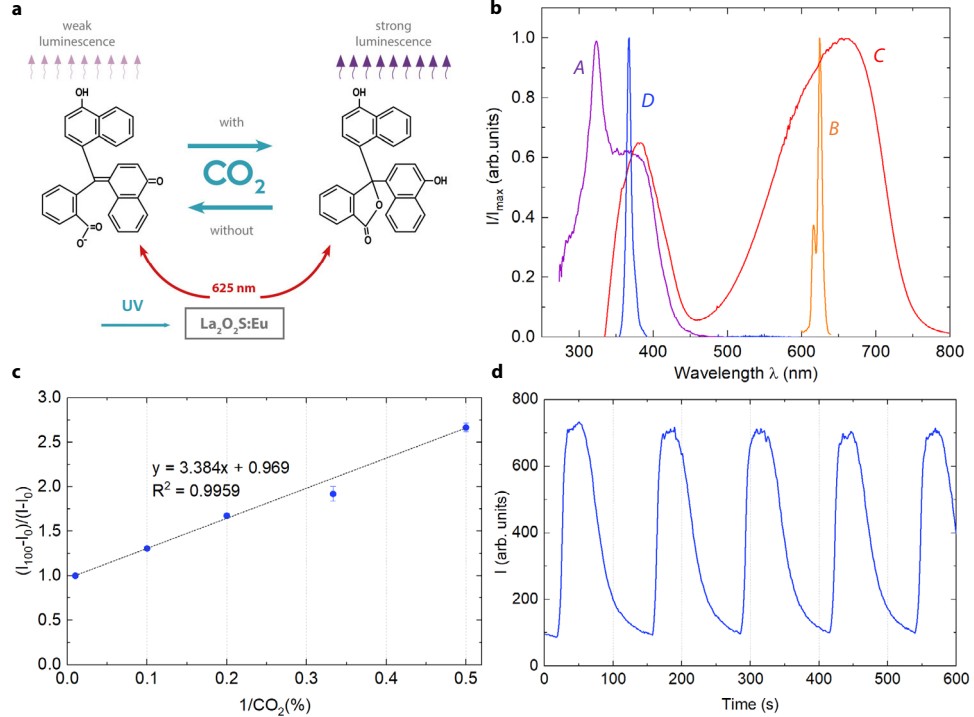

**Fig. 2 $CO_2$ sensing scheme and characterization. a** $CO_2$ sensing mechanism. **b** Spectral properties of the constituents of the $CO_2$ sensor: (A) Absorption spectrum of $La_2O_2S$:Eu; (B) Emission spectrum of $La_2O_2S$:Eu; (C) Absorption spectrum of the basic form of α-naphtholphthalein; and (D) Emission spectrum of the UV LED. **c** Relationship between the luminescence intensity and the $CO_2$ concentration fitted to the equation by Nakamura and Amao[25] ($n = 5$. Data are presented as mean values ± 1SD). **d** Response and recovery time of the developed sensor while increasing and decreasing $CO_2$ concentrations from 0% to 100% and vice versa.

time from 10% and 90% of the maximum signal of 9 s and a recovery time from 90% to 10% of 30 s were obtained.

**Design and performance of the sensing NFC flexible tag.** Figure 3a shows the circuit block diagram of the system and its interconnections. The conditioning circuitry comprises as main electronic components an extreme low-power microcontroller unit (XLP MCU) for control and data processing purposes; an NFC Dynamic Tag IC along with a custom-designed NFC antenna for NFC link and power management; and a sensing module consisting of a UV LED, a digital colour sensor and a temperature sensor. Additional details on the selection criteria of every component based on the system requirements can be found in Supplementary Section 3. The sensing module is placed aligned facing the $CO_2$-sensitive membrane, as shown in Fig. 3b. The selected UV LED has a peak emission at 367 nm to optically excite the $CO_2$-sensitive membrane, whose luminescent response is detected by the digital colour sensor, which filters out UV excitation light and is sent to the MCU for further processing. A temperature sensor was also included in the design to correct the temperature dependence of the chemical sensor. In this application, the power consumption of the whole system is 4.45 mW in operation mode, which is roughly 20% of the total available power that can be sourced from the chosen NFC chip, i.e. 22.5 mW. Low-power electronic design and component selection has made this possible, thus enabling battery-free operation of the system.

The designed tag was printed on a 125 μm-thick PET flexible substrate. PET film was chosen because of the combination of three key properties required for our development: (i) Its high optical transmittance of ~90%, needed to avoid the attenuation of the optical sensor signal; (ii) Its high flexibility, which allows bending angles beyond those needed for the given application;

and (iii) Its excellent adherence and high melting point (~270 °C), which are crucial characteristics for the successful deposition and thermal curing of the conductive Ag-based ink. Photographs of the fabricated flexible tag are shown in Fig. 3c–e, with final dimensions of $60 \times 45$ mm². Tag size was designed as a result of the compromise between achieving enough inductive coupling while making the system as compact as possible by placing the circuitry within the inner space of the planar coil. Considering the cost of the different parts comprising the sensing tag, where the ICs are the most expensive components, the estimated large-scale production cost of the whole sensing system is below 5€ (see Supplementary Table 4). As for the sensing membrane, the cost can be estimated below 0.01€ without considering mass production. The thicker component of the design is the MCU, so the thickness of the tag is 1.75 mm in the worst of the cases. When placed on the inner face of a facemask as shown in Fig. 1b, the tag is completely unnoticeable to the user thanks to its reduced thickness and compact size, making it comfortable to wear within the facemask. Furthermore, the presence of the PET tag does not significantly affect the gas permeability of the facemask, only increasing the $CO_2$ concentration in the DSV by a relative 5% compared to the tagless facemask.

As depicted in Fig. 4a, the final designed antenna coil antenna had $N = 5$ turns and dimensions of 43 mm × 58 mm, being 400 μm the width of the conductor and 600 μm the interspacing between the lines. Figure 4b–d shows the surface topology of the printed flexible tag. More detailed information can be found in Supplementary Sections 3–5.

NFC tags typically have a reading distance between 1 and 5 cm when scanned with a smartphone. In practice, the range is very variable and dependent on the tag antenna size and design, tag quality, the IC used and the reader antenna design, among others[26]. The decrease of the inductive coupling with the antenna

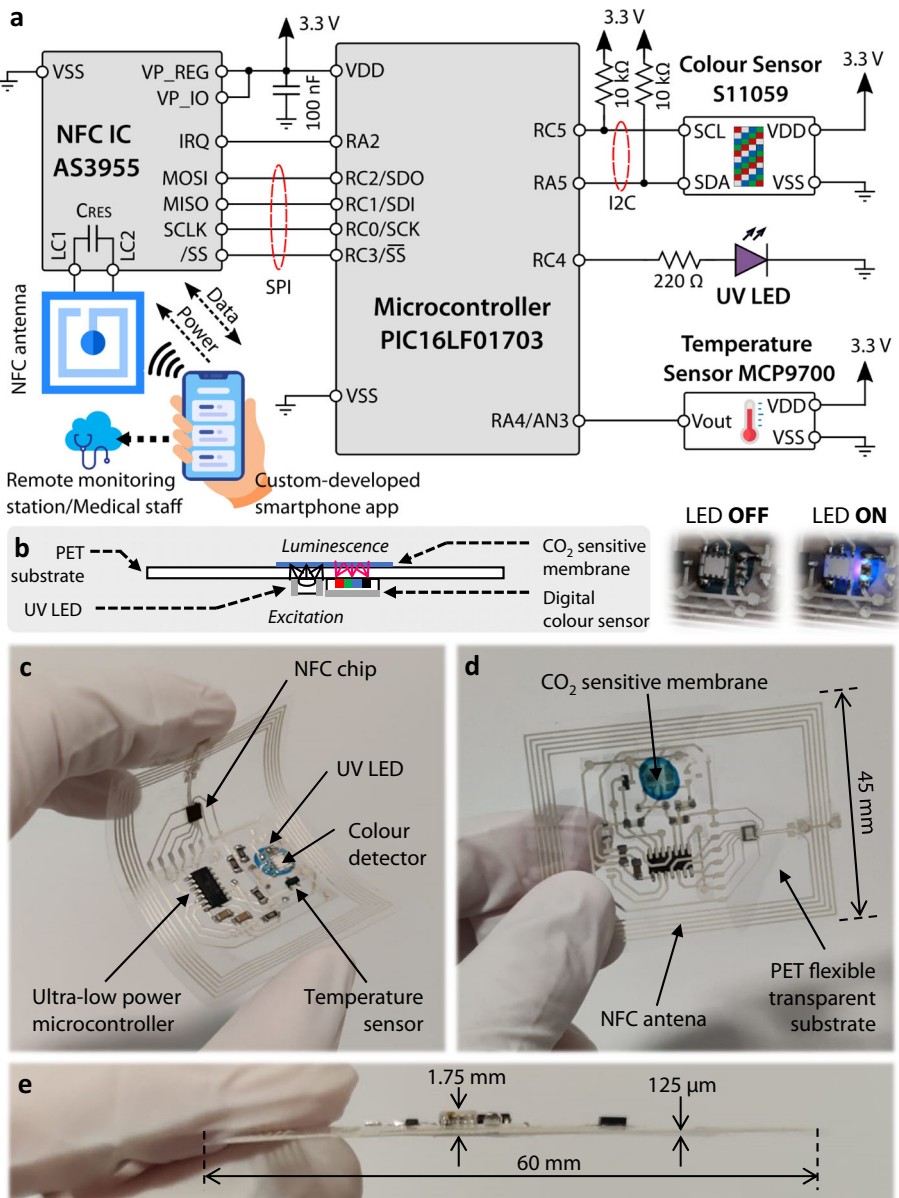

**Fig. 3 NFC flexible tag design. a** Circuit block diagram of the developed NFC tag for wireless $CO_2$ readout. **b** Schematic diagram of the sensing procedure in which the UV LED and colour sensor is placed in front of the $CO_2$ sensitive membrane. UV excitation and red luminescence can be seen in the picture on the right with LED ON. **c–e** Photographs of the flexible tag printed on the 125 μm-thick PET substrate.

separation was also computed through the coupling constant, defined as $k = M_{12}/\sqrt{L_1 \cdot L_2}$, where $L_1$ and $L_2$ are the inductance of the smartphone and our system coils, respectively. In Fig. 4e, the normalized coupling constant ($k/k_0$) is depicted as a function of the coil separation, showing the decrease of the mutual inductance. $k_0$ is the maximum coupling constant between coils when aligned. In our case, with the flexible tag in a flat position and using the Samsung Galaxy S7 smartphone as the NFC reader (Samsung Group, Seoul, South Korea), we obtained an experimental maximum separation of 19 mm before the platform turned off. Therefore, about 25% of maximum coupling is required to power up our system, resulting in certain freedom when it comes to the antenna positioning. The magnetic flux density and flux lines are depicted as an inset in Fig. 4e with a separation of 19 mm between both coils.

Further, since the developed system is flexible, different curvatures were tested to study their impact on the tag's operation and reading distance. As shown in Fig. 4f, the sensing system operates correctly with a reading distance above 10 mm up to a bending angle of ~110° along both bending axes. This bending value is beyond the required ones for the given application since the tag inside the facemask lies virtually flat. However, there might be occasions where some bending is needed depending on the shape of the user's face. Even so, we estimate maximum bending angles in practice between ~20° and 30°. Therefore, the tag would still be readable with sufficient range for the given application in all cases. Nevertheless, it is worth mentioning that due to the attachment of rigid ICs to the flexible substrate, the physical integrity of the system might be affected by improper use of the facemask in terms of excessive bending, twisting or crunching.

**Smart facemask calibration and characterization.** As mentioned, the NFC-based tag measures the opto-chemical sensor luminescence through the RGB colour coordinates provided by

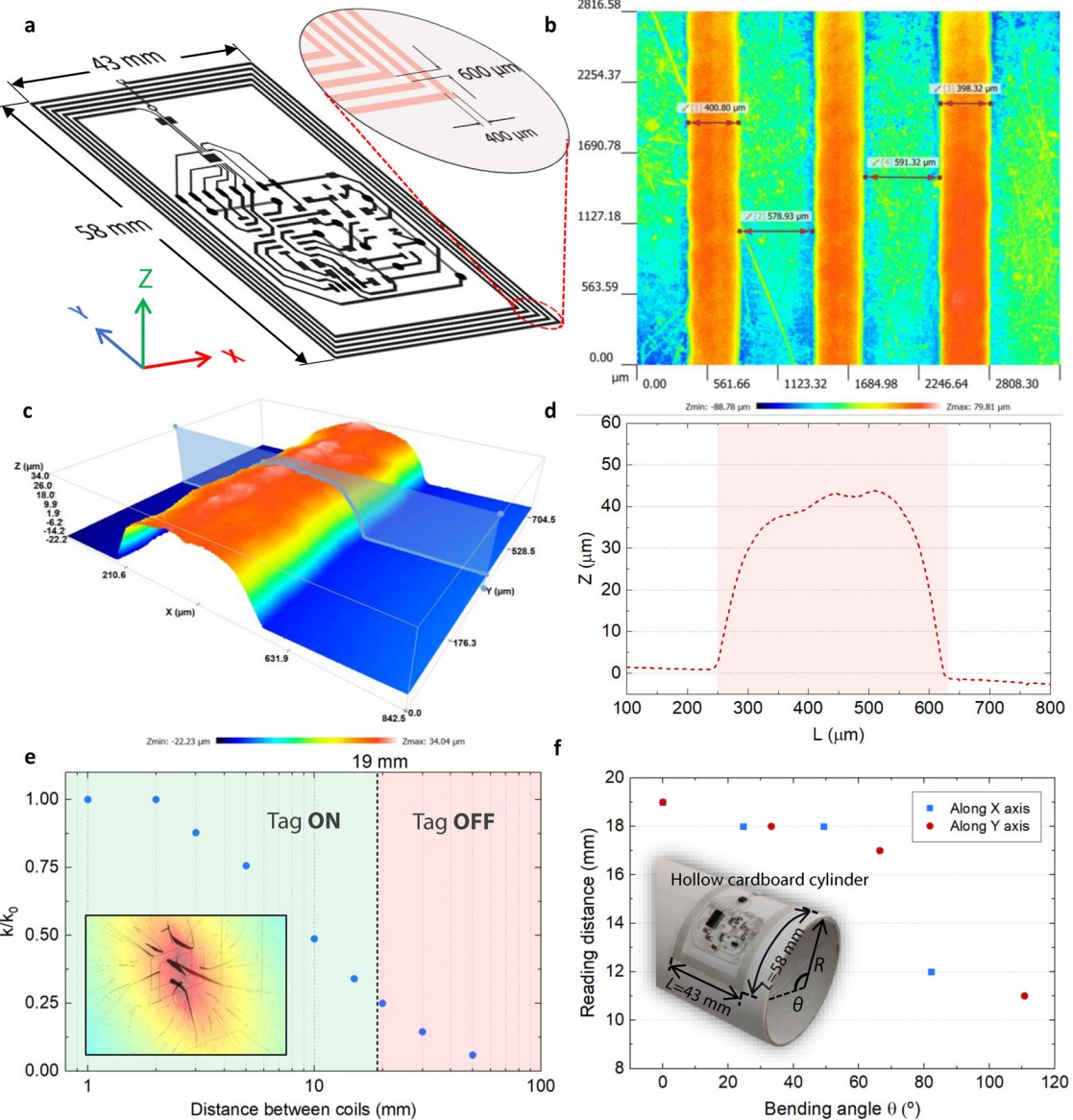

**Fig. 4 NFC flexible tag characterization. a** Dimension of the designed planar coil acting as NFC antenna. **b**, **c** 3D surface images of the printed flexible tag measured by the optical profilometer. **d** Example of the 2D profile of one inductor track showing the printed track height. **e** Normalized coupling factor as a function of the distance between coils numerically calculated, showing the separation intervals where the tag is ON or OFF. **f** Reading distance as a function of the bending angle.

the digital colour detector. We have found that the $R$ coordinate was the most sensitive parameter to the $CO_2$ concentration. Therefore, once that the theoretical chemical sensor response was already verified by Eq. (1), a simpler relationship between the $R$ coordinate and the gas concentration was proposed. Thus, the calibration curve actually programmed in the sensing tag is shown in Fig. 5a, where the relationship between the red component ($R$) of the luminescence intensity and $CO_2$ concentrations, ranging from 500 to 48,000 ppm, is depicted. The $R$-value is normalized to the maximum value reached in the calibration curve, which corresponds to the higher concentration of $CO_2$. Five replicas were taken at room temperature (25 °C) for each concentration. It can be seen from the fitting curve that the relation is modelled using a second-grade equation ($[CO_2] = aR^2 - bR + c$), being $a$, $b$ and $c$ fitting parameters, in this case $a = 1.76 \times 10^5$, $b = 1.5 \times 10^5$ and $c = 2.7 \times 10^4$ ($r^2 = 0.994$). From this equation, it is clear that an increase in the $CO_2$ concentration leads to an increment in the $R$ coordinate. Although the standard deviations of the measurements are included in the figure, the

error bars are unnoticed. The resolution of the system can be derived from the fitting function[27]. The value obtained in the worst case (at 4.8%) is 221 ppm, and in the range of interest, the resolution achieved is 103 ppm, which is very close to the resolution provided by the manufacturer of the reference instrument described in the section below (100 ppm). Moreover, response and recovery times are below 1 s in this $CO_2$ concentration range.

The dependencies of the $R$ colour coordinate of the $CO_2$ sensing membrane luminescence with temperature and relative humidity (RH) were also monitored using a thermostatic chamber VCL4006 (Vötsch Industrietechnik, Germany). It was proved in previous works that temperature has a noticeable influence on the quenching produced by different gases in the emitted intensity[28]. For this system, the temperature drift caused in the $R$ coordinate at room conditions can be observed in Fig. 5b. This effect has been compensated using the temperature sensor included in the tag design: the measured temperature value is provided for each $R$-value, so a compensated value of $R$ is

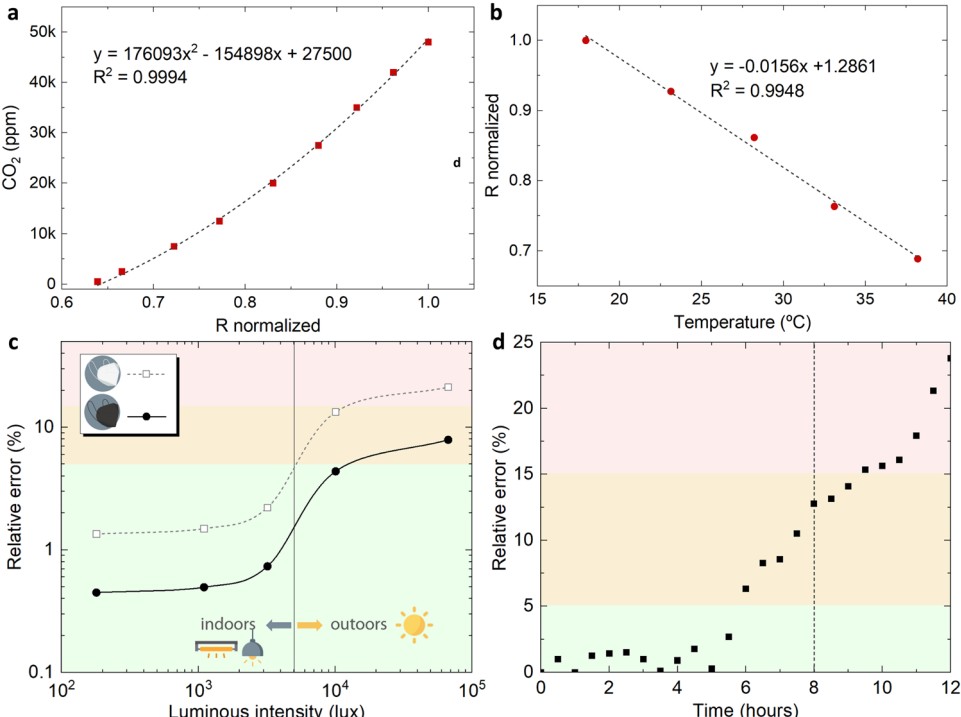

**Fig. 5 Smart facemask calibration and characterization. a** Calibration curve and **b** temperature dependence of the $CO_2$ sensing membrane ($n = 3$. Data are presented as mean values ±1 SD. Error bars are on the order of graph point size). **c** Ambient light influence: Reading relative error under different ambient light conditions (both indoors and outdoors) using the sensing system inside white and black FFP2 facemasks. The lines connecting the experimental data points are plotted as eye guides. **d** Sensor lifetime: Reading relative error over a continuous 12-h period of time replicating the conditions inside an FFP2 facemask.

obtained using the corresponding fitting curve (Fig. 5b) to get a corrected prediction of $CO_2$ concentration. For this purpose, a linear temperature coefficient of $\alpha = (-15.6 \pm 0.7) \times 10^{-3}\,K^{-1}$ is used, as experimentally obtained. Besides, a humidity sweep study was conducted at 30 °C in the range from 10% to 90% RH. The obtained results indicate that the $R$ coordinate of the sensing membrane is independent of the humidity in the ambient air. This is a very remarkable result since it eliminates the need for monitoring RH inside the facemask, which is around 70% on average under normal circumstances.

Due to the nature of the sensing mechanism (i.e. based on optical measurements), environmental light can interfere with the system compromising the system performance. To assess the effect of ambient light on the system functionality, the sensing tag was placed inside an FFP2 facemask worn by the user under different ambient light conditions. Figure 5c shows the reading relative errors obtained as a function of the luminous intensity. Three regions have been plotted as a visual guide: green zone for relative errors below 5%; orange for relative error values between 5% and 15%; and red zone for relative errors above 15%. The vertical line in the graph differentiates between two regions, corresponding to indoor and outdoor light conditions. The graph shows the differences between the white and black facemasks with regard to optical shielding. As expected, black facemasks deliver lower error values since they achieve better light shielding for the optical measurements than white facemasks. Predictably, the errors increase considerably outdoors, where only black face masks should be used to keep the errors below 8% even in direct sunlight. In the case of indoor environments, also white facemasks provide enough light shielding to keep measurement errors below 3% for all cases, while black facemasks deliver errors below 1% in indoor scenarios.

Regarding chemical sensor aging, caused by typical dyes photooxidation, sensor lifetime was estimated by analysing how

long the sensor is able to operate under intensive working conditions inside the facemask. Figure 5d shows the relative error of $CO_2$ concentration over a continuous 12-h period of time. It can be observed how the measurements are very stable (relative error < 3%) during the first 5 ½ h. From then on, the relative error starts increasing. When it reaches 8 h of usage, the relative error is ~13%. This error could be considered as a maximum allowable error. Therefore, we could assume a sensor lifetime of at least 8 h, which is the same as the FFP2: NR (not reusable) facemask usage time recommended by manufacturers. At that moment, the costless chemical sensor should be replaced after the corresponding cleaning and disinfection of the tag (e.g. by means of UV radiation), since the electronic tag itself is reusable.

**Smart facemask performance.** The performance of the smart facemask was preliminarily assessed for both short-term and long-term use in static conditions (i.e. sitting and working on a computer) as depicted in Fig. 6a, b, respectively. In the first case, the $CO_2$ concentration was monitored every 30 s through the smart facemask system as well as using a reference instrument as the gold standard (MultiRAE Lite, RAE Systems, Inc. by Honeywell, CA, USA) to compare and evaluate the results. This is a wireless, portable multi-gas monitor of up to six gases. In the case of the $CO_2$ sensor, it has a measurement range up to 50,000 ppm with a resolution of 100 ppm. Figure 6a shows how both curves are in very good agreement. The variation of the responses of both sensors can be explained in terms of the different gas sampling procedures. The reference system was set to pump gas inside the sensor chamber for 10 s before each measurement in order to provide stable readings. On the other hand, the facemask sensor response lays on the absorption/desorption balance of $CO_2$ inside the sensing membrane at every time. Therefore, in the

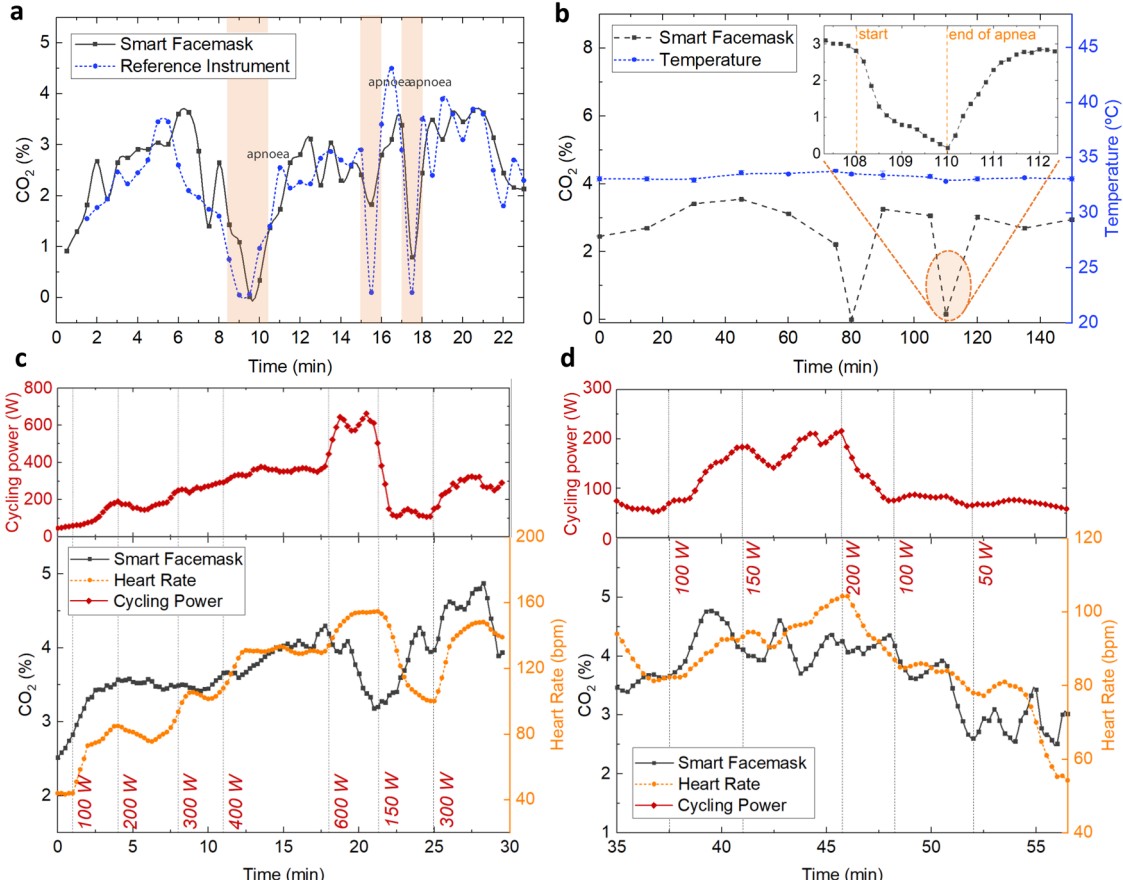

**Fig. 6 Smart facemask performance.** Study of the smart facemask performance (*n* = 1) during **a** short-term use; **b** long-term use; and **c, d** graded cycling exercise tests.

former, an integrated measurement is obtained but assigned to one time instant, whereas more instantaneous data is gathered in the latter. In any case, both systems showed very similar trends beyond punctual differences. It can be observed that during the first minutes of the test, the results delivered by the developed system are slightly delayed with respect to the ones given by the reference instrument. This is due to the time needed to fill the mask's DSV. Three apnoeas were done by holding the breath during the test, in which both instruments (smart facemask and reference) responded accordingly, as shown in Fig. 6a. In the case of the long-term test, the facemask was worn for 2.5 h and the measurements were taken every 15 min. Temperature values were also monitored through the temperature sensor included in the NFC tag. Figure 6b shows the stability of the measurements taken during this test, where the obtained $CO_2$ concentration was around 3% with the exception of two apnoeas that were done at minutes 80 and 110. During the last one, a more exhaustive measurement protocol was conducted by taking readings every 10 s. The inset graph included in Fig. 6b shows how the system allows a very clear reconstruction of the $CO_2$ reduction during the two-minute apnoea (min 108 to 110) and the subsequent $CO_2$ increment from minute 110 of the test.

The performance of the developed smart facemask was also assessed during a graded cycling exercise test by monitoring the data delivered by the instrumented facemask as well as the cycling power and heart rate (HR). Figure 6c, d shows the 4-period Simple Moving Average (SMA) of the calculated $CO_2$ concentration and HR as the cycling power varies during the cycling exercise test. Both figures also include the SMA power during the exercise as delivered by the cycling simulator software. This

parameter is closely correlated to the HR trend (see red and orange lines in Fig. 6c, d). These tests showed that the concentration of $CO_2$ in the breathing zone while wearing the facemask during the exercise increased by ~1.9%, which is in line with other studies found in the literature[15,29]. Interestingly, this experiment also revealed a time-delayed correlation between the cycling power and the measured $CO_2$ concentration by the smart facemask, as well as with the HR during the exercise, as observed from Fig. 5c, d[4,5]. In fact, we have estimated that the HR response was delayed 3 min with respect to $CO_2$ variations during this experiment. Although this interesting result is beyond the scope of our paper, it may trigger further investigations and research in this field. Even though the conducted performance tests do not constitute a formal clinical trial, their purpose is to give an idea of the potential of the developed system in the field of wearable electronics for non-invasive health monitoring.

## Discussion

Since the global pandemic declaration by the WHO due to the spread of COVID-19, the universal use of facemasks has been widely recommended or even mandatory in many public or workspaces for a large number of individuals and long durations as a means of source control. Therefore, the integration of sensors in conventional facemasks to monitor human physiological and behavioural signals can open new pathways in the field of wearable healthcare devices[30]. In this context, a number of interesting developments of sensing facemasks can be found in the literature, as summarized in Table 1. Our system distinguishes from previous solutions by combining the following three main aspects simultaneously: (i) Both the sensor and the electronics are

**Table 1 Comparison of sensing smart facemasks that can be found in the literature.**

| Measured parameter | Transmission method | Powering method | Substrate material | | Electronics | Reference |
|---|---|---|---|---|---|---|
| | | | Sensor(s) | | | |
| Ammonia, formaldehyde, & ethanol gases | Visual LED indicator | Battery | Flexible nylon rope | | Rigid PCB | 40 |
| Particulate matter (PM) sensor | Wired or wireless (Bluetooth) | Battery | Commercial device (rigid) | | Rigid PCB | 41 |
| Temperature & strain | Wireless (N/S) | Battery | Flexible PI | | Rigid PCB (Arduino) | 42 |
| Humidity | Wireless (UHF RFID) | Energy harvesting (UHF RFID reader) | Flexible PI | | Flexible PI and PET | 43 |
| Moisture | Wireless (UHF RFID) | Energy harvesting (UHF RFID reader) | Woven fabric | | Woven fabric | 44 |
| HR/BP, SpO$_2$ & temperature | Wireless (Bluetooth) | Battery | Commercial sensors (rigid) | | Rigid PCB | 45 |
| Respiratory rate | Digital LCD screen | Energy harvesting (Electret nanogenerator) | PEI electret nonwoven | | N/A | 46 |
| O$_2$, CO$_2$ & activity (accelerometer) | Wireless (Bluetooth) | Battery | Commercial sensors (rigid) | | Rigid PCB | 47 |
| CO$_2$ & temperature | Wireless (HF RFID: NFC) | Energy harvesting (NFC-enabled smartphone) | Flexible PET | | Flexible PET | This work |

integrated on a flexible substrate, thus achieving better conformability and wearability; (ii) It is able to determine CO$_2$ concentration, a key parameter during the process of breathing that has been hardly ever contemplated in the current state-of-the-art of sensing facemasks; and (iii) It implements wireless powering through RF energy harvesting and data transmission using NFC technology, thus making the system compatible with any NFC-enabled device without modification, including smartphones, through a custom-developed application. As can be seen in Table 1, some other wireless solutions have been implemented in previous smart facemasks designs, such as UHF radio frequency identification (RFID) or Bluetooth. In our case, NFC was chosen over Bluetooth or other wireless protocols because it has a key advantage: its ability to simultaneously transfer data and, more importantly, power between devices offers the opportunity of battery-less communication, as the required energy to supply the system can be obtained from the electromagnetic field generated by the remote reader. This means that NFC eliminates the requirement of including a battery in the tag, with all the benefits that it entails in terms of less environmental impact, lower weight, lower cost, less maintenance, and increased wearability and comfort. Additionally, the choice of NFC technology over other protocols such as UHF RFID allows the use of any NFC-enabled smartphone as the remote reader instead of an ad-hoc reader, making this technology within the reach of virtually any user and thus enabling self-health management. Another advantage of NFC as compared to other technologies relies on the possibility to produce NFC tags by establishing high-volume manufacturing methods (for example, roll-to-roll) on flexible substrates such as PET or paper. Besides, the short range of NFC can be seen as an advantage to maintain secure communication and avoid access to the data by other devices in the vicinity.

Apart from the functionalities offered by the developed smartphone application, additional features could be easily implemented to endow the system with extra options. As an example, further data processing of the measured parameters can be included to accommodate the warning levels to different scenarios depending on the final user requirements or application. Global positioning system (GPS) location function can be also easily added to keep track of the user's routine and possibly establish more accurate early warnings. Automatic or scheduled report shipping to a remote monitoring station or medical staff could also be implemented through the already existing sharing option. All these abilities, combined with the wearable and wireless features of the system, make this powerful tool a strong candidate for adoption in numerous areas such as non-invasive monitoring of physical and sport activity, at the workplace, or for preclinical research, prognostics and diagnostics of illnesses (e.g. in chronic obstructive pulmonary disease (COPD) patients) together with other medical equipment.

The ability to perform a real-time readout of CO$_2$ concentration inside a facemask by a single wearable device in a non-invasive way is important for a number of reasons. Several studies have revealed that wearing a facemask alters the air exchange by extending the physiological DSV of the respiratory system by the dead-space volume of the facemask itself[29]. This increases the volume of the previous exhalation (rich in CO$_2$ and depleted in O$_2$), which is re-inhaled with each breath[31]. In the reviewed studies, dead space CO$_2$ accumulation during common conditions ranged from 1.5% (e.g. at rest, during the speech, or at low work rates) up to ~3.5% (during low-intensity exercise), which is well beyond the expected in atmospheric air (~0.04%)[9,32]. Our results are in line with the reviewed literature, with CO$_2$ values around 2% during low work-rate activities (Fig. 6a, b) and peak values up to ~5% during high-intensity exercise (Fig. 6c, d). Several studies in the literature have also been devoted to

investigating the impact of wearing facemasks during physical exercise[13,15,17,33]. Some studies have related prolonged $CO_2$ rebreathing to symptoms of discomfort, fatigue, dyspnoea, dizziness, headache, sweating, muscular weakness and drowsiness, even in individuals without medical diseases[5,9,32,34]. Even though in healthy populations, the potentially life-saving benefits of wearing facemasks seem to outweigh the documented adverse effects[35], there seems to be widespread agreement on the increased breathing resistance, $CO_2$ rebreathing and decreased inhaled $O_2$ concentration caused by the use of facemasks[18,32]. In any case, the potential negative effects of $CO_2$ are known to be related to the duration of exposure, apart from the gas concentration itself. As an example, the guidelines of some regulatory organizations recommend a threshold limit value of 0.5% $CO_2$ in the workplace atmosphere (averaged over an 8-h workday), or consider a 30-min exposure to 4% $CO_2$ in the ambient atmosphere as dangerous for health[36]. By definition, respiratory acidosis (i.e. a condition occurring when the lungs cannot remove enough of the produced $CO_2$ by the body) can arise in a healthy individual with a moderate physical load when exposed to a 10% $CO_2$ concentration for at least 30 min[37–39]. In this context, we propose a non-invasive system that delivers information to the user not only about the measured $CO_2$ concentration inside the facemask but also in terms of exposure time to such concentration. With this information, the user is advised to ventilate the facemask under particular scenarios through the accompanying smartphone application.

In summary, we have developed a platform for gaseous $CO_2$ determination consisting of a flexible, battery-less, near-field-enabled sensing tag located inside an FFP2 facemask. Both the sensor and the conditioning circuitry have been fully characterized and evaluated, and examples of performance during daily activity and exercise monitoring demonstrate wireless powering and sensing in unconstrained environments. All these features strengthen the potential applications of the proposed device in the fields of non-invasive health monitoring, preclinical research, prognostics and diagnostics with wearable electronics. Furthermore, the versatility of the system could be the object of future research since it is possible to combine several optical chemical sensors in a single flexible passive NFC tag[27] to determine several gases concentrations of interest for health status.

## Methods

**Materials and reagents.** Hydroxypropyl methylcellulose (HPMC, Methocel E-5, LV USP/EP premium grade) (Dow Chemical Iberia S.L., Tarragona, Spain) was used as the membrane polymer. The reagents tetramethylammonium hydroxide pentahydrate (TMAOH) as the phase transfer agent; Tween 20, cetyltrimethylammonium bromide (CTAB), sodium dodecyl sulfate (SDS) as surfactants; α-naphtholphthalein as pH indicator; 1-ethyl-3-methylimidazolium tetrafluoroborate (EMIM BF4) as ionic liquid, all sourced from Sigma-Aldrich Merck (Madrid, Spain). Lanthanum oxysulfide: Europium ($La_2O_2S$:Eu) as luminophore, from Phosphor Technologies (Stevenage, UK). Sheets of PET from Goodfellow (Cambridge, UK) were used as the support for the membranes. In order to characterize the sensing membranes, the standard mixtures are prepared using $N_2$ as the diluting gas, controlling the flow rates of the high purity $CO_2$ and $N_2$ gases (Linde Gas España, Spain) entering the mixing chamber with computer-controlled mass flow controllers. The system worked at a flow rate of 500 $cm^3 min^{-1}$ and total pressure of 760 Torr.

**$CO_2$ sensor design and fabrication.** The cocktail for $CO_2$ sensing membrane was prepared by adding 1 ml of a solution of HPMC 1% w/w in water to 450 μL of a solution of 6.6 mg α-naphtholphthalein in ethanol, 5 μL of Tween 20, 2 μL of EMIM BF4, 50 μL of TMAOH $6.04 \times 10^{-2}$ M and 3 mg of $La_2O_2S$:Eu. The cocktail was deposited on the PET substrate placed on an H6-23 spin-coater (Laurell Technologies Corporation, North Wales, PA, USA) rotating at an initial speed of 80 rpm for 60 s and then rising to 180 rpm until it dried. To characterize the optical response of the $CO_2$ sensing membranes, Cary Eclipse Varian Inc. (Palo Alto, CA, USA) luminescence spectrometer steady-state luminescence measurements were used.

**Flexible tag fabrication and characterization.** The designed tag was printed on a 125 μm-thick PET flexible substrate using a Voltera V-One PCB printer (Voltera Inc., Ontario, Canada) by means of a five-step process. Firstly, the circuit layout was printed on the PET substrate using flexible conductive Ag-based ink (Flex 2, SKU: 1000383, Voltera Inc.). After the circuit was printed, the ink was immediately cured using the heated platform available at the Voltera V-One printer at 160 °C for 15 min. Afterward, the solder paste was dispensed on the components' footprints and via pads (T5-Sn42 Bi57.6 Ag0.4, SKU: 1000356, Voltera Inc.). Subsequently, the electronic components and the bridges were manually placed on top of their footprints and, finally, a reflowing phase was conducted on the hotplate. The reflow profile consisted of a soak temperature of 140 °C for 45 s followed by a peak temperature of 160 °C for 30 s.

The initial NFC antenna inductance design was estimated through the Grover Method (see details in Supplementary Section 4), considering the space requirements for the electronic components in the inner space of the coil as shown in Fig. 3c, d. After the initial estimation, the final dimensions and number of turns for our planar coil were obtained through the optimization process with numerical simulation using advanced design simulator (ADS, Keysight Technologies, Santa Clara, CA, USA) and COMSOL Multiphysics (COMSOL Group, Stockholm, Sweden). Both 2D profiles and 3D surface images were obtained using an S Neox 3D Optical Profiler (Sensofar, Barcelona, Spain) and SensoVIEW 1.6 software. The frequency response of the printed antenna was evaluated using a Precision Impedance Analyzer (see details in Supplementary Section 5).

**Bending tests.** To conduct the reading distance and bending tests, the tag was bent by fitting it to the convex surfaces of hollow cardboard cylinders with different radii ($R$), namely $R = 100$, 50 and 30 mm. In such scenarios, the bending angle ($\theta$) in radians can be calculated as the component's length in the curvature direction ($L$) divided by the cylinder radius ($R$). Two cases were considered for each radius, depending on if the tag was bent along the X-axis ($L = 43$ mm) or along the Y-axis ($L = 58$ mm). In the first case, for the selected radii the bending angles ranged from ~25° up to ~82°, while for the second case the bending angles varied from ~33° up to ~110°. According to the conducted experiments, the tag reading distance was reduced with increasing bending radius. When the tag was bent along the X-axis, the reading distance only decreased to 18 mm (5.2% decrease) for $R = 100$ and 50 mm, while it went down to 12 mm (36.8% decrease) for the case of $R = 30$ mm. If the bending line was along the Y-axis, the reading distance reduced to 18 mm (5.2% decrease), 17 mm (10.5% decrease) and 11 mm (42.1% decrease) for the three bending radii, respectively.

**Smartphone application development.** Android Studio 4.1.3 was used as the integrated development environment (IDE) to code the application, which was designed and tested against API level 26 (Android 8.0). However, it supports different Android versions as the lowest API level compatible with the application is API 18 (Android 4.3). The developed smartphone application and the microcontroller firmware are included as a Supplementary Software file.

**Ambient light tests.** Five different scenarios were tested with illuminance levels ranging from ~200 lx up to ~67 klx, which was quantified using a lux meter RS Pro LX-105 (RS Components, London, UK). To achieve the different light conditions up to ~3.5 klx, a dimmable LED ring light (ELEGIANT EGL-02S) was employed, while for the two last conditions (>10 klx), the tests were conducted outdoors in the shadow and under direct sunlight, respectively. For each scenario, the average of three measurements of the $R$ coordinate was taken and the relative error for each case was calculated from the measurements with and without illumination (in the dark, i.e. at 2 lx). Two different colours of standard FFP2 facemasks were tested: white and black. In all cases, $CO_2$ concentration during the experiments was virtually constant at ~2.4%, which is a common value during low work-rate activities.

**Sensor lifetime test.** To replicate the conditions inside a facemask worn by a user while controlling such conditions at the same time, the sensing system was placed inside the climate chamber VCL4006 (Vötsch Industrietechnik, Germany) at $T = 30$ °C and RH of 70%. An atmosphere of constant 2% $CO_2$ was created inside the chamber. Using this setup, we can make sure that the differences in the $CO_2$ readings are only caused by the sensor aging over time, and not due to some other external conditions or changes in the $CO_2$ concentration. Three measurements of the $R$ coordinate and temperature were taken every 30 min during 12 h (when sensor aging became unsuitable for further usage). The smartphone was placed inside the chamber and it was remotely controlled via USB using an external PC.

**Exercise test setup.** The study was conducted according to the guidelines of the Declaration of Helsinki, and approved by the Ethics Committee in Human Research (CEIH) of the University of Granada (2190/CEIH/2021). Informed consent was obtained from subjects involved in the study. The subject under test was a 43-year-old healthy male (77 kg in weight and 181 cm in height) in a very good physical shape. A smart bicycle roller and realistic cycling simulator (BKOOL Cycling Pro, Valencia, Spain) were used for the cycling exercise test. With this tool, a bicycle is paired to a simulator software and the user can check in real-time the values of power, cadence, speed, etc. while pedalling. The heart rate (HR) in beats

per minute (bpm) was monitored simultaneously using a Polar H10 HR sensor chest strap Bluetooth paired with a Polar M400 sport watch (Polar Electro Ltd., Kempele, Finland). R 3.6.1 and Microsoft Excel 2019 were used for data analysis.

**Statistics and reproducibility**. No statistical method was used to predetermine sample size. No data were excluded from the analyses. The experiments were not randomized. The Investigators were not blinded to allocation during experiments and outcome assessment. R 3.6.1 and Microsoft Excel 2019 were used for data analysis.

**Reporting summary**. Further information on research design is available in the Nature Research Reporting Summary linked to this article.

## Data availability
All relevant data supporting the key findings of this study are available within the article and its Supplementary Information files or from the corresponding author upon reasonable request. Source data generated in this study are provided in the Source Data file. Source data are provided with this paper.

## Code availability
Source codes for microcontroller firmware (developed with MPLAB X IDE v5.45) and Android™ smartphone application (SmartMask v1.0) are available at an open-access repository (URI: http://hdl.handle.net/10481/71668) under a Creative Commons license.

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

## Acknowledgements

This study was funded by Spanish MCIN/AEI/10.13039/501100011033/ (Projects PID2019-103938RB-I00 and ECQ2018-004937-P) and Junta de Andalucía (Projects B-FQM-243-UGR18, P18-RT-2961 and postdoctoral grant of PE DOC_00520). The projects were partially supported by European Regional Development Funds (ERDF).

## Author contributions

Conceptualization, M.D.F.-R., A.M.O., A.J.P., L.F.C.-V.; methodology, P.E., N.L.-R., O.M.-R., I.M.P.V., M.A.C.; software, P.E., N.L.-R., A.M.O.; evaluation, P.E., O.M.-R., N.L.-R., A.M.O., I.M.P.V, M.A.C.; investigation, M.D.F.-R., P.E., A.M.O., A.J.P., L.F.C.-V.; writing—original draft preparation, P.E., M.D.F.-R.; writing—review and editing, all authors; funding acquisition, A.J.P., L.F.C.-V. All authors have read and agreed to the published version of the manuscript.

## Competing interests

The authors declare no competing interests.
