## [Peer Review File · Nature Communications]

Reviewers' Comments:

Reviewer #1:

Remarks to the Author:

The manuscript "Smart facemask for wireless CO₂ optical monitorization through flexible printed battery-less NFC tag" by P. Escobedo et al. presents a facemask with a smart battery free flexible hybrid integrated sensing system for CO₂ detection".

The work is interesting, showing noteworthy results and significance to the interdisciplinary field of wearable electronics for health status monitoring. Conclusions and claims are well supported. Methodology and data interpretation are both sound, meeting expected standards in the field. There are nevertheless several open issues and questions which require major and mandatory revisions:

- the title is misleading, as the whole system is not printed but rather hybrid considering the use of Si chips. I would suggest to adapt the title to better describe the work.
- both abstract and introduction would benefit from some more quantitative data on the new developed solution.
- introduction on the topic and the work needs improvement, e.g. it is not clear what new paradigm is created by smart facemasks and what these add to the non smart ones?
- there are several aspects of the whole system realization which are not clearly justified. In particular, firstly it is not clearly explained why NFC was chosen as compared to other protocols, considering the short range of communication requiring un-wearing of the mask for both charging and communication. Secondly, the choice of the sensing mechanism is not clear as compared to other - more simple and real-time - technologies available. Considerations of aspects such as power consumption, overall timings (time of sensor use vs time of system charge, etc) and use of bulky and thick (1.75 mm) ICs have also not been fully justified. Finally, re-usability and cost need to be also addressed.
- a comparison of the realized CO₂ sensor with other CO₂ sensors is surely required for all relevant sensor performance parameters.
- please compare the results achieved with the printed tag with SoA printed antennas and NFC tags.
- the real usage of the smart mask in terms of bending, twisting, crunching etc should be better investigated. In particular bending should be provided also in terms of strain applied and the variations upon bending will also need to be better explained.
- the sensor variation as compared to the results with the reference sensor will also need to be better explained.
- it would be useful to outlook in more details how the system could be expanded in terms of more measurements and physiological parameters.
- please check all numbering of figures as some typos have been found, e.g. at line 297-298 the text refers to the wrong figures in the supplementary information.

Reviewer #2:

Remarks to the Author:

This manuscript depicted a smart mask sensing platform for gaseous CO₂ monitoring with near-field-enabled data and power transmission. Although wearable electronics especially smart mask got tremendous attention due to need for tele-healthcare, the CO₂ monitoring in this project is questionable. First, because the gas exchange of FFP2 facemasks with filtering facepiece capacity is very poor, they are only recommended for special work situations rather than daily wear. For daily protection during the epidemic, even medical staff only need to wear medical masks. Hence, the use of monitoring CO₂ in FFP2 masks seems not obvious for epidemic devices. Second concern is about "real-time monitoring". The system needs to actively provide power through an external NFC chip of external equipment such as a smartphone. Otherwise, it can't transmit data in real time. It's not really "real-time monitoring". it is just wireless data transmission. Thirdly, the effect on gas permeability of mask itself needs to be assessed due to the final dimensions (60×45 mm²) of the device based on PET is it takes up a quarter of the mask area. Fourthly, in practical application, the red coordinate by photodetector is susceptible to the ambient light. The accuracy

of sensors under different ambient light is uncertain. Finally, authors need to pay attention to the difference between LOD and sensitivity, as well highlight the value of the developed chip as a health monitoring system by pointing out whether the results can be used to prognosis the health status. Overall, this manuscript is limited in material innovation or advanced technological breakthrough, and thus is not recommended for publication in Nature Communications.

Key results

The paper describes a new concept for CO₂ sensing in face masks. The system is built up completely with many different aspects of the engineering development (sensor, electronic hard- and software, communication) as well as physiological experiments being described. This completeness of the development approach makes the paper somehow unique. The CO₂ sensor and its measurements are tested in practical examples with good correlations demonstrating the practical usage potential of such system.

Validity, Significance

The data and descriptions taken for and from the system are well presented and the fact that the system practically works speak for itself in terms of the validity of the approach. Some statistical and long-term data i.e. for the sensor, its cross-sensitivities and drifts are lacking, but only to an extent which goes beyond an initial prototype study (which the reviewer here considers not to be very relevant for the quality of the paper).

The individual parts of the system described in the paper are not very well compared to the state of the art: while this is a difficult task and needs specifications to be benchmarked i.e. at the interfaces of the individual block/tasks described here, this would help readers to better understand significance and new aspect specific to this work. The authors lack from making clear what they do different here and why compared to other approaches for CO₂ sensing, for building NFC powered systems, and for their physiological studies (all of this could be put into respect to the application).

Data and methodology

The authors present enough data demonstrating the principle and its general suitability as a face-mask indicator for high CO₂ concentration there. It is on the one hand side a matter of the fact of the completeness of the problem they try to solve, that many aspects of the individual parts and its suitability in the system are missing, but on the other hand side it is not fully clear to the reviewer whether the experimental data taken here is sufficient to prove the feasibility of such approach in a broader context (e.g. different people, different micro- and macro-climate, ...). Therefore, in the 'suggested improvements' section a few suggestions are made.

Analytical approach, clarity and context

As mentioned above, a clear indication and discussion on the how and why certain approaches have been taken here is missing. This unfortunately reduces the impact of the paper with respect to education of readers and new ongoing research in this area.

Some parts of the work are rather well explained (even beyond the level of newness) like the Android interface, however in other parts of the text more detail may be of higher scientific interest (see 'suggested improvements').

Suggested improvements

Below you can find a list of suggestions and items to improve with priority according to the listing:

- General structure: the NFC system as shown here and its electrical behavior is very close to industry standards and commercial hardware in many aspects. The parts used here can be bought commercially even at higher integration level for such sensors (e.g. Texas Instruments RF430FR215x). However, the integration on flexible substrates and its consequences for the circuits (e.g. may cause strain changes at the device level with impact on electrical behavior)

as well as the approach to measure and package the sensor are really new and are only described here on a very low level of detail (see later suggestions)

- It is not clear to the reviewer whether the NFC powering and exposure to a magnetic field causes side effects to the CO₂ measurement accuracy – and if not: how this is avoided?
- The same question arises with stray light – here an answer is given, but the reviewer would also be interested in the effects of air humidity in this system and its correlation the CO₂ measurement principle

Putting less emphasis on the standard parts of the system and more on how to optimize for the purpose would most likely be appreciated by most readers.

- In a similar context, more details about how the wavelength planning for good sensitivity in this system, the optimization of the solution and the packaging of the sensor material are done would be appreciated as well ...

- The measurement of the CO₂ takes rather long (9s – as stated in table 1). For an NFC powered system using a cell phone as a power source next to it this appears to be rather long to the reviewer. The reviewer however assumes the measurement does not need all power of the system all time.

Why not using a small battery back-up via a super-capacitor or thin film battery to enhance the measurement time with respect to the read time? (e.g. the Texas Instrument chip mentioned above e.g. has power management support capabilities for such systems and this concept is used in commercial sensor system like blood sugar measurement (Abbott))?

- The android app programmed here is needed for the system demonstration, however the general structure of the firmware is rather straightforward, while corrections for drifts, sensor calibration, or time delay are not detailed well. This (or maybe the authors know even better examples of novelty in the context of this application) however, is what the reader of this Journal wants to know!?
- Figure 5:
 - In subfigure c, the signal from the face mask sensor seems to somehow integrate the other sensors CO₂ measurement values – is this correct, or are these statistical outliers?
 - It is maybe beyond the expertise of the authors (and it is definitely beyond the expertise of the reviewer), but the time delayed correlation between cycling power and CO₂ as well as heart rate seems to be an interesting result which may trigger further investigation and also could lead toward new research aspects?
- Two minor issues:
 - Line 328: What does HR mean here? – later it is defined as ‘Heart Rate’ (line 374), but this does not make sense here!?
 - Line 93: typo ‘showed’ -> ‘shown’

● Reviewer #1

The manuscript "Smart facemask for wireless CO₂ optical monitorization through flexible printed battery-less NFC tag" by P. Escobedo et al. presents a facemask with a smart battery free flexible hybrid integrated sensing system for CO₂ detection. The work is interesting, showing noteworthy results and significance to the interdisciplinary field of wearable electronics for health status monitoring. Conclusions and claims are well supported. Methodology and data interpretation are both sound, meeting expected standards in the field.

Response: We thank the reviewer for appreciating the work accomplished and for their valuable revision. Below are addressed all the comments point by point.

There are nevertheless several open issues and questions which require major and mandatory revisions:

- The title is misleading, as the whole system is not printed but rather hybrid considering the use of Si chips. I would suggest to adapt the title to better describe the work.

Response: We agree with the reviewer on this observation related to the hybrid nature of the system, in which rigid ICs are attached to a flexible substrate with printed interconnects. To avoid any confusion in this matter, we have removed the word 'printed' from the title, which now reads:

"Smart facemask for wireless CO₂ optical determination through flexible battery-less NFC tag"

- Both abstract and introduction would benefit from some more quantitative data on the new developed solution.

Response: We appreciate the reviewer's suggestion and agree that both these sections will benefit from including some quantitative information. Therefore, the following text has been added in the revised version of the **Introduction** section:

"The proposed sensing system is based on a novel and stable opto-chemical sensor with a limit of detection of 140 ppm of CO₂ concentration and resolution of 103 ppm for measuring this gas concentration in the DSV of a facemask. Response and recovery times are below 1 s in this CO₂ concentration range, while sensor lifetime is 8 hours, comparable with FFP2 facemask recommended usage time. Both the sensor and the conditioning circuitry have been fabricated onto a light and flexible polymeric substrate conforming what is usually named as sensing tag with final dimensions of 60×45 mm²."

Abstract has also been rewritten to include some more quantitative data:

"Main features are resolution and limit of detection of 103 and 140 ppm respectively, and sensor lifetime of 8 h, comparable with recommended FFP2 facemask usage time"

- Introduction on the topic and the work needs improvement, e.g. it is not clear what new paradigm is created by smart facemasks and what these add to the non-smart ones?

Response: As stated by the reviewer, this sentence may lead to confusion. What we intended to mean was that this smart facemask development can be included into the *Internet of Things* (IoT) paradigm. In that way, the sentence has been rewritten at the end of the first paragraph of the **Introduction** section:

“While standard facemasks simply act as air filters for the nasal and/or mouth passage, the integration of sensors to control parameters of interest can be an added value to improve their usage and effectiveness, including the developed smart facemask into the wearable devices of the Internet of Things (IoT) paradigm.”

- There are several aspects of the whole system realization which are not clearly justified. In particular, firstly it is not clearly explained why NFC was chosen as compared to other protocols, considering the short range of communication requiring un-wearing of the mask for both charging and communication.

Response: We thank the reviewer for this comment, which helped us to improve several aspects of the system justification and provide rationale for our methodological choices. Firstly, NFC was chosen because it fills important technological gaps that cannot be sufficiently addressed with other wireless technologies (such as other RFID bands, Bluetooth or Wi-Fi). In particular, its ability to simultaneously transfer data and, more importantly, power between devices offers the opportunity to design battery-free sensing systems. This means that NFC eliminates the requirement of including a battery in the tag, with all the benefits that it entails in terms of less environmental impact, lower weight, lower cost, less maintenance, and increased wearability and comfort. Additionally, the choice of NFC technology enables the use of any NFC-enabled smartphone as the remote reader instead of an ad-hoc reader, making this technology within the reach of virtually any user. Another advantage of NFC as compared to other technologies relies on the possibility to produce NFC tags by established high-volume manufacturing methods (for example, roll-to-roll) on flexible substrates such as polyethylene terephthalate (PET) or paper. Some of these advantages of NFC over other technologies were already mentioned in the first paragraph of the **Discussion** section, while others have been included to clearly justify why NFC was chosen:

“As can be seen in Table 2, some other wireless solutions have been implemented in previous smart facemasks designs, such as UHF Radio Frequency Identification (RFID) or Bluetooth. In our case, NFC was chosen over Bluetooth or other wireless protocols because it has a key advantage: its ability to simultaneously transfer data and, more importantly, power between devices offers the opportunity of battery-less communication, as the required energy to supply the system can be obtained from the electromagnetic field generated by the remote reader. This means that NFC eliminates the requirement of including a battery in the tag, with all the benefits that it entails in terms of less environmental impact, lower weight, lower cost, less maintenance, and increased wearability and comfort. Additionally, the choice of NFC technology over other protocols such as UHF RFID allows the use of any NFC-enabled smartphone as the remote reader instead of an ad-hoc reader, making this technology within the reach of virtually any user and thus enabling self-health management. Another advantage of NFC as compared to other technologies relies on the possibility to produce NFC tags by established high-volume manufacturing methods (for example, roll-to-roll) on flexible substrates such as PET or paper. Besides, the short range of NFC can be seen as an advantage to maintain secure communication and avoid the access to the data by other devices in the vicinity.”

Finally, it is important to highlight that the short-range communication required by the NFC protocol to enable wireless inductive coupling does not in any way imply the un-wearing of the mask for charging and/or communicating. On the contrary, both powering and communication between the reader (i.e., the smartphone) and the sensing tag are –and must be– conducted while wearing the mask. The measurement process can be observed in the *Supplementary Movie* (from

minute 1:20 onwards). Besides, the short range of NFC can be seen as an advantage to maintain secure communication and avoid the access to the data by other devices in the vicinity.

- Secondly, the choice of the sensing mechanism is not clear as compared to other - more simple and real-time - technologies available.

Response: We appreciate these valuable comments. A detailed rationale for the choice of the sensing mechanism has been added to first paragraph of the *CO₂ sensing scheme* section:

“Existing methods for CO₂ sensing can be classified into four techniques: infrared absorptiometry, catalytic bead sensors, electrochemical devices, and optical sensors. Regarding infrared sensors, they are subject to more strong interference as well as an increase in the cost and the size of the final prototype due to the characteristics of available sensors. The catalytic bead sensors are based on flameless combustion, and the electrochemical sensors, although effective and reliable, can be cross-sensitive to other gases. Due to the requirement of low-power consumption of the proposed application, these techniques were discarded. Optical sensors (see a brief revision in Supplementary Table 1), based on the luminescence measurement of a very stable inorganic phosphorus modulated by an acid-base indicator depending on the CO₂ concentration, have been widely studied showing advantages that are really attractive for the developed system: possibility of miniaturization, electrical isolation and less sensitivity to noise. ”

Moreover, we have included a State-of-the-Art brief revision in *Supplementary Section 1* (see Supplementary Table 1). A rationale for the sensor composition has been exposed as well:

“As a State-of-the-Art brief revision, Supplementary Table 1 summarizes the principal analytical characteristics of representative CO₂ luminescent gas sensors found in the literature. CO₂ detection chemistry for optical sensors is generally based on plastic solid-state sensor film that works with a colour-based or fluorescence-based pH indicator, such as α -naphtholphthalein or 8-hydroxypyrene-1,3,6-trisulfonate with pKa that matches CO₂ acidity. The limitation of this design arises from the poor stability of the membranes. Due to the small number of existing fluorescent indicators, a combination of luminescent dye with non-luminescent pH indicator has been used. This system works by both primary or secondary inner-filter effect or by resonance energy transfer (RET), measuring the intensity of the luminescence, the decay time, ratiometric signals or colour. The drawback of these strategies is the limited shelf life that arises from photobleaching of the dye, evaporation of its trace water content, and degradation of the quaternary amine hydroxide base. Different solutions have been tested, such as the inclusion of ionic liquids, or bases that do not suffer from Hoffman degradation. Another possible strategy in sensors based on secondary inner-filters is to replace the luminescent dye, whose emission is modulated by the pH indicator, by an LED with the same wavelength of the dye. Another approach developed to improve the stability of sensor membrane is to replace the luminescent dye, typically a metal complex or an organic molecule, by stable inorganic phosphors. This last solution has been adopted in this study, in which the luminescence emission of an inorganic phosphor is absorbed by the basic form of the pH indicator, thus increasing the luminescence when the CO₂ concentration increases.

Regarding the approach to measure and package the sensor, the strategy used is based on the displacement of the acid-base equilibrium of a pH indicator immobilized on the membrane in the presence of CO₂. To increase the sensitivity of the sensor towards CO₂, an inert luminescent dye, whose emission band overlaps the absorption band of the pH indicator in its basic form, is co-immobilized on the membrane. In absence of CO₂, the entire indicator will be in basic form due

to the presence of a quaternary ammonium hydroxide. The luminescence of the dye will be completely cancelled by a secondary internal filter mechanism. In the presence of CO₂, there will be a shift towards the acid form of the indicator with the consequent increase in luminescence. A stable inorganic phosphor is selected as luminescent material, which increases the stability and lifetime of the membrane. Additionally, an ionic liquid is included in the membrane composition to strengthen the sensor stability and sensitivity by increasing the solubility of CO₂ in the membrane. In addition, a surfactant is included to reduce the interfacial tension and facilitate the permeation of CO₂ gas in the membrane. All these components are encapsulated in a hydrophilic hydroxypropylmethylcellulose membrane to prevent water loss over the life of the membrane.

Supplementary Table 1. Comparison of luminescence-based sensors for CO₂ gas determination found in the literature.

Sensing chemistry	Technique	LOD (ppm)/ range (%)	Response time (s)	Precision (%)	Storage lifetime (months)	Ref.
HPTS/TBP/TOAOH	LI	0-1%	4.3	-	24-36 h	1
HPTS/TOAOH/sol gel/TiO₂	LI-A/DLR	80	50	-	7 days/in vacuum	2
Ru(dpp)/Sudan III/TOAOH	LI/LT	60	20-30	1.6	3/dark	3
PAACHl	FI	2	20-30	-	12/4-8°C	4
PtOEP/N/TOAOH	FI	0-40%	60	-	14 days/dark	5
NaYF₄:Yb/BTB/TBAO H	FI	1100	10	1%	-	6
GAB/aB/IL	I	2600	23	0.70	At least 120 days/ dark	7
Cr/ETH2412/TDDMA Cl	A	200	-	CI	-	8
La₂O₃S:Eu/N/TMAOH/ C	C	140	9	4.41	140 days/dark	This work

HPTS: 8-hydroxypyrene-1,3,6-trisulfonate; TBP: Tributylphosphate; TOAOH: Tetrabutylammonium hydroxide; Ru-dpp): Ruthenium (II) tris (4,7-diphenyl-1,10-phenanthroline); PAACHl: Chlorophyllide b derivatized polyallylamine; PtOEP: Platinum octaethylporphyrin; N: α -Naphtholphthalein; BTB: Bromothymol blue; TBAOH: Tetrabutylammonium hydroxide; GAB: Cr(III)-doped gadolinium aluminium borate; IL: Ionic liquid; aB: AzaBodipy; Cl: Carbonate ionophore; TDDMACl: Tridodecylmethylammonium chloride; TMAOH: Tetramethylammonium hydroxide; LI: Luminescent intensity, A: Absorbance, R: Ratiometric, LT: Luminescent lifetime; DLR: Dual luminophore reference; C: Colour.

- Considerations of aspects such as power consumption, overall timings (time of sensor use vs time of system charge, etc) and use of bulky and thick (1.75 mm) ICs have also not been fully justified.

Response: Regarding other aspects such as power consumption, the energy harvesting capabilities of the chosen NFC chip allow a regulated supply power up to 22.5 mW according to the manufacturer. Bearing this in mind, the electronic design was optimized for low-power operation, resulting in a power consumption of the whole system of only 4.45 mW in operation mode. Therefore, the power consumption requirement was successfully accomplished to enable a

battery-free design. This information has been now added to *Design and performance of the sensing NFC flexible tag* section:

“In this application, the power consumption of the whole system is 4.45 mW in operation mode, which is roughly 20% of the total available power that can be sourced from the chosen NFC chip, i.e. 22.5 mW. Low-power electronic design and component selection has made this possible, thus enabling battery-free operation of the system.”

With regard to timings, we would like to point out that the system does not need any charge time. As it was specified in the first paragraph of *Smart facemask calibration* section of the original manuscript, response and recovery times are below 1 s in the CO₂ concentration range of interest (up to 5 %):

“Moreover, response and recovery times are below 1 s in this CO₂ concentration range.”

As an example, *Supplementary Movie* shows from minute 1:20 onwards how the measurement is taken virtually immediately upon approach of the smartphone to the sensing tag.

Finally, the use of packaged ICs is unavoidable at the current stage of the proof-of-concept prototype. In the literature, circuits referred to as flexible printed circuits (FPCBs) usually include conventional packaged rigid ICs soldered to a flexible substrate. As an example, in the following paper¹ the authors present a wireless, NFC-based, battery-free sensing system which they denominate as ‘flexible’ even if it contains several rigid components soldered to the flexible substrate:

Heo SY, Kim J, Gutruf P, Banks A, Wei P, Pielak R, Balooch G, Shi Y, Araki H, Rollo D, Gaede C, Patel M, Kwak JW, Peña-Alcántara AE, Lee KT, Yun Y, Robinson JK, Xu S, Rogers JA. Wireless, battery-free, flexible, miniaturized dosimeters monitor exposure to solar radiation and to light for phototherapy. Sci Transl Med. 2018;10(470):eaau1643. doi: [10.1126/scitranslmed.aau1643](https://doi.org/10.1126/scitranslmed.aau1643)

When it comes to an industrial level, one possible solution is to use bare dies, a strategy that has been employed for years in commercial RFID/NFC tags. These have a very small (<1 mm²) IC, the RFID/NFC chip, attached at two points to the antenna. The ICs are so small that their flexibility is almost irrelevant at any realistic bending radius. However, as the ICs become more capable (for example those required for a sensing system such as microcontroller, memory, etc.) and hence larger, difficulties begin to arise. In this context, flexible ICs are beginning to emerge by thinning Si chips to achieve thicknesses as low as 10 μm. These fragile dies are then encapsulated in a thin layer of polyimide to add fracture protection. In principle, this methodology is applicable to many semiconducting chips (e.g., the NFC chip and the microcontroller unit used in our system). Finally, the other alternative is the natively flexible ICs, but at present they are only available for relatively simple applications. Figure R1 shows photographs of the described ICs for their comparison.

Figure R1. Top: Packaged IC (left), bare die (centre) thinned die (right); Bottom: Natively flexible ICs [Source: <https://www.idtechex.com/en/research-article/how-flexible-integrated-circuits-unlock-their-potential/20717>]

In any case, all these solutions are still an emerging technology frontier for complex sensing systems², which goes beyond the initial prototype proposed in this work. Anyway, the justification for the use of each electronic component in the sensing system has been included throughout the revised manuscript:

- **NFC chip:** The power consumption of the system was 4.45 mW in operation mode. Even if this can be considered as low-power consumption, not all the available NFC ICs in the market are able to provide that amount of energy at the required regulated voltage level and current³⁻⁵. As an example, there are some other solutions combining NFC chip and microcontroller unit in the same IC (e.g. RF430FRL15xH), but they are not able to provide the required power for the given application. The employed NFC chip was chosen because its energy harvesting capabilities allow a regulated supply power up to 22.5 mW (up to 5 mA @4.5V). This information has been now added to ***Design and performance of the sensing NFC flexible tag*** section:

“In this application, the power consumption of the whole system is 4.45 mW in operation mode, which is roughly 20% of the total available power that can be sourced from the chosen NFC chip, i.e. 22.5 mW. Low-power electronic design and component selection has made this possible, thus enabling battery-free operation of the system.”

- **Microcontroller Unit (MCU):** In line with the above, the MCU was chosen due to its eXtreme Low-Power (XLP family) features, low operating voltage range (1.8V to 3.6V) and digital/analog peripherals: serial communication (I²C and SPI), and 10-Bit Analog-to-Digital Converter (ADC). In fact, the MCU usage has been optimized in a way that all its available pins are used (see Fig. 3a). These criteria have been now included in the first paragraph of ***Supplementary Section Sensing NFC tag components***:

“The MCU model was chosen due to its eXtreme Low-Power (XLP family) features, low operating voltage range (1.8 V to 3.6 V) and digital/analog peripherals: serial communication (I²C and SPI), and 10-Bit Analog-to-Digital Converter (ADC). All those features were required to communicate with the NFC chip through the SPI port; with the digital colour sensor through the I²C port; and to get the temperature readings by means of the ADC module. In fact, the MCU usage has been optimized in a way that all its available pins are used (see Fig. 3a).”

- **Digital colour detector:** This part was chosen because of a combination of several reasons: low voltage operation (2.5 V or 3.3 V), which was compatible with our requirements; low current consumption (75 μ A typical); it supports the I²C interface, which was necessary to communicate with the MCU; and because its sensitivity to red emission (centered at $\lambda=615$ nm) overlaps the luminescent emission spectrum of La₂O₂S:Eu for the sensing mechanism (see Fig. 2b). The information in the first paragraph of *Supplementary Section Sensing NFC tag components* has been now expanded to include more details:

“The digital colour sensor used in this design is the S11059-02DT (Hamamatsu Photonics, Hamamatsu, Japan), which is an I²C interface-compatible digital detector sensitive to red ($\lambda_{peak} = 615$ nm), green ($\lambda_{peak} = 530$ nm), blue ($\lambda_{peak} = 460$ nm), and near infrared ($\lambda_{peak} = 855$ nm) incident radiation, which is codified in the corresponding four 16-bit digital words. Its sensitivity to red emission (centered at $\lambda=615$ nm) overlaps the luminescent emission spectrum of La₂O₂S:Eu for the sensing mechanism (see Fig. 2b). Besides its high resolution, this detector was selected due to its low voltage operation (2.5 V or 3.3 V) and low power consumption of only 250 μ W in operation mode, which makes it optimal for our battery-less design.”

- **LED:** The exciting LED was selected on the basis of the absorption spectrum of the La₂O₂S:Eu phosphor, which is shown in Fig. 2b. It can be seen that the maximum peak occurs at 325 nm, so ideally the LED should have its emission spectra at this wavelength to optimize the sensing mechanism. Unfortunately, we have not been able to find any commercially available LED at that particular frequency that, additionally, meets our design requirements regarding low polarization voltage, low-power consumption and small size. The chosen LED model that meets such requirements has its emission spectra at 367 nm, which is not the optimum value but still overlaps a valid band of the absorption spectrum of the La₂O₂S:Eu phosphor, as can be seen in Fig. 2b. This explanation has been now included in *Supplementary Section Sensing NFC tag components*:

“The UV LED was selected on the basis of the absorption spectrum of the La₂O₂S:Eu phosphor, in which the maximum peak occurs at 325 nm (see Fig. 2b). Therefore, the LED was selected to have its peak emission as close as possible to this wavelength to optically excite the CO₂ sensitive membrane, whilst meeting our design requirements in terms of low polarization voltage, low-power consumption and small size. To the best of the authors’ knowledge, the selected LED model with peak emission at 367 nm was the optimum one meeting the design requirements as far as possible. Once excited by the UV LED, the luminescent response delivered by the CO₂ sensitive membrane is detected by the digital colour sensor and sent to the MCU for further processing.”

- **Temperature sensor:** The temperature sensor model MCP9700A was used to correct the temperature drifts of the chemical sensor. This model was chosen because of its miniature size (2.90×1.30×0.95 mm³) and because it presents an accuracy of ± 1 °C and a very low current consumption of only 6 μ A. This justification is now included in *Supplementary Section Sensing NFC tag components*:

“A temperature sensor model MCP9700A (Microchip Technology Inc.) was also included in the design, connected to one 10-bit Analog to Digital Converter (ADC) input of the MCU, to correct the temperature drifts of the chemical sensor. This sensor was chosen due to its miniature size (2.90×1.30×0.95 mm³) and because it presents an accuracy of ± 1 °C and a very low current consumption of only 6 μ A.”

- Finally, re-usability and cost need to be also addressed.

Response: We agree with the reviewer that these are relevant matters. Firstly, re-usability has been addressed by conducting new experimentation in order to analyse how long the sensor is able to operate under intensive working conditions inside the facemask. This has been called sensor lifetime, and is different from the sensor storage lifetime (see Table 1 and Supplementary Section 2) since storage and operating conditions are not the same. To replicate the conditions inside a facemask worn by a user while controlling such conditions at the same time, the sensing system was placed inside the climatic chamber at $T=30\text{ }^{\circ}\text{C}$ and relative humidity of 70%. An atmosphere of constant 2% CO_2 was created inside the chamber. Using this setup, we can make sure that the differences in the CO_2 readings are only caused by the sensor aging over the time, and not due to some other external conditions or changes in the CO_2 concentration. Three measurements of the R coordinate and temperature were taken every 30 min during 12 hours (when sensor aging became unsuitable for further usage, as explained below). The smartphone was placed inside the chamber and it was remotely controlled via USB using an external PC. It is worth mentioning that we are conducting an aging experiment under intensive conditions, which are not usually given in a practical application since it is not usual to continuously wear a facemask during 12 hours.

Fig. 5d shows the relative error of CO_2 concentration over the 12-hour period of time. This relative error has been computed as $\Delta[\text{CO}_2]/[\text{CO}_2]_{t=0}$, where $\Delta[\text{CO}_2]$ is the difference between the measured CO_2 concentration at any given time and $[\text{CO}_2]_{t=0}$, which is the first reading at $t = 0\text{ h}$, considered as the reference value. It can be observed how the measurements are very stable (relative error $< 3\%$) during the first $5\frac{1}{2}$ hours. From then on, the relative error starts increasing. When it reaches the 8 hours of usage, the relative error is $\sim 13\%$, which means that instead of reading a CO_2 concentration value of 2%, the delivered value is $\sim 2.25\%$. This error could be considered as a maximum allowable error, so we can assume a sensor lifetime of at least 8 h, which is the same as the FFP2: NR (not reusable) facemask usage time recommended by manufacturers. At that moment, the costless chemical sensor should be replaced on the tag after the corresponding cleaning and disinfection (e.g. by means of UV radiation), since the electronic tag itself is reusable.

We have included the description of the sensor aging experiment in the **Materials and Methods** section:

“Sensor lifetime test. To replicate the conditions inside a facemask worn by a user while controlling such conditions at the same time, the sensing system was placed inside the climatic chamber at $T=30\text{ }^{\circ}\text{C}$ and relative humidity of 70%. An atmosphere of constant 2 % CO_2 was created inside the chamber. Using this setup, we can make sure that the differences in the CO_2 readings are only caused by the sensor aging over the time, and not due to some other external conditions or changes in the CO_2 concentration. Three measurements of the R coordinate and temperature were taken every 30 min during 12 hours (when sensor aging became unsuitable for further usage). The smartphone was placed inside the chamber and it was remotely controlled via USB using an external PC.”

New figure (Fig. 5d) and discussion has been added to the **Smart facemask calibration and characterization** section:

“Regarding chemical sensor aging, caused by typical dyes photooxidation, sensor lifetime was estimated by analysing how long the sensor is able to operate under intensive working conditions inside the facemask. Fig. 5d shows the relative error of CO_2 concentration over a continuous 12-hour period of time. It can be observed how the measurements are very stable (relative error $<$

3 %) during the first 5 ½ hours. From then on, the relative error starts increasing. When it reaches the 8 hours of usage, the relative error is ~13 %. This error could be considered as a maximum allowable error. Therefore, we could assume a sensor lifetime of at least 8 h, which is the same as the FFP2: NR (not reusable) facemask usage time recommended by manufacturers. At that moment, the costless chemical sensor should be replaced after the corresponding cleaning and disinfection of the tag (e.g. by means of UV radiation), since the electronic tag itself is reusable.”

Figure 5. Smart facemask calibration and characterization. *a* Calibration curve and *b* temperature dependence of the CO₂ sensing membrane. *c* Ambient light influence: Reading relative error under different ambient light conditions (both indoors and outdoors) using the sensing system inside white and black FFP2 facemasks. The lines connecting the experimental data points are plotted as eye guides. *d* Sensor lifetime: Reading relative error over a continuous 12-hour period of time replicating the conditions inside a FFP2 facemask.

Apart from this, the cost of the developed sensing system has been estimated considering the cost of its different components, where the ICs are the most expensive ones. In this way, we estimate a large-scale production cost below 5 € for the whole sensing system. This information has been included in the first paragraph of **Design and performance of the sensing NFC flexible tag** section of the revised manuscript:

“Considering the cost of the different parts comprising the sensing tag, where the ICs are the most expensive components, the estimated large-scale production cost of the whole sensing system is below 5 €. As for the sensing membrane, the cost can be estimated below 0.01 € without considering mass production.”

- A comparison of the realized CO₂ sensor with other CO₂ sensors is surely required for all relevant sensor performance parameters.

Response: The new table (*Supplementary Table 1*) presented above shows some representative optical sensors found in the literature and their main parameters. The realized CO₂ sensor stands out compared to other sensors in terms of increased life time, sufficient sensitivity to reach atmospheric levels and an extremely low cost per membrane.

- Please compare the results achieved with the printed tag with SoA printed antennas and NFC tags.

Response: We have included in the revised manuscript (*Supplementary Section NFC antenna characterization*) the following discussion comparing the results achieved with the State-of-the-Art printed NFC antennas:

“Performance of NFC antennas are primarily determined by their size, quality factor (Q), capability of energy harvesting (or magnetic coupling factor), and read range. Moreover, antenna size and shape is of paramount importance for the capability of energy harvesting. It has been shown that this feature can be maximized when both the receiver and the transmitter antennas have equal dimensions¹³. This factor was firstly considered for our NFC antenna design, given the NFC antenna of the smartphone employed in this work. Taking advantage of this size, the rest of the NFC tag components were located in the inner space within the innermost antenna turn for the sake of compactness. This design strategy has been previously reported in biomedical applications^{14,15}. Regarding the tag power consumption, which is 4.5 mW in our case, is comparable to other similar designs, ranging from 1.3 mW¹⁶ to 6.5 mW¹⁷. In fact, according to the datasheet the selected NFC chip can provide up to 22.5 mW in optimal condition, which is higher than other NFC ICs such as SL13A (around 12 mW), MLX90129 (15 mW), and M24LR04E-R (18 mW). On the other hand, if we compare the measured Q factor = 15.15, which meets the Q factor standards of commercial coils at this resonance frequency (i.e. 13.56 MHz), is comparable to those reported previously by Rogers’ group ($Q \sim 14$)¹⁸ and others¹³. Finally, the obtained read range, between 2 and 1 cm depending on the bending conditions, is also aligned with previous reports¹⁹. To sum up, the designed printed antenna presents features that are comparable to previous high quality NFC sensing solutions, using an affordable and compact printer compared to bulkier ink-jet and screen printers.”

- The real usage of the smart mask in terms of bending, twisting, crunching etc should be better investigated. In particular bending should be provided also in terms of strain applied and the variations upon bending will also need to be better explained.

Response: We appreciate the reviewer’s insightful suggestion and understand that the flexible characteristic of the system can lead to request further experimentation on this matter; however, on one side, we unfortunately do not have the required instrumentation to carry out strain measurements and, on the other side, we believe that the conducted analysis in terms of bending is enough for the given application. As stated in the last paragraph of *Design and performance of the sensing NFC flexible tag* section, the studied bending curvatures up to $\sim 110^\circ$ are beyond the maximum bending angles that the facemasks go through under normal and proper use (estimated at $\sim 20^\circ$ - 30°). It is worth pointing out that the conducted analysis of the reading distance as a function of the bending angle (see Fig. 4f) also involves the correct operation and functionality of the whole sensing system in such scenarios. For these reasons, we believe that further investigation on the flexibility of the system is neither feasible, nor would significantly add a differential value since such an analysis is beyond the scope of our paper, which aims only to

provide a proof-of-concept prototype. Nevertheless, we recognize that the physical integrity of the system might be affected by incorrect and improper use of the facemask in terms of excessive bending, twisting or crunching, so this limitation has been mentioned in the paper by adding the following sentence in the last paragraph of the aforementioned section:

“Further, since the developed system is flexible, different curvatures were tested to study their impact on the tag’s operation and reading distance. As shown in Fig. 4f, the sensing system operates correctly with a reading distance above 10 mm up to a bending angle of $\sim 110^\circ$ along both bending axes. This bending value is beyond the required ones for the given application, since the tag inside the facemask lies virtually flat. However, there might be occasions where some bending is needed depending on the shape of the user face. Even so, we estimate maximum bending angles in practice between $\sim 20^\circ$ - 30° . Therefore, the tag would still be readable with sufficient range for the given application in all cases. Nevertheless, it is worth mentioning that due to the attachment of rigid ICs to the flexible substrate, the physical integrity of the system might be affected by improper use of the facemask in terms of excessive bending, twisting or crunching.”

- The sensor variation as compared to the results with the reference sensor will also need to be better explained.

Response: The variation of the responses of both sensors (Fig 5c) can be explained in terms of the different gas sampling procedures. Reference system was set to pump gas inside the sensor chamber during 10 s before each measurement in order to provide stable readings. On the other hand, facemask sensor response lays on the absorption/desorption balance of CO₂ inside the sensing membrane at the time. Therefore, in the former an integrated measurement is obtained but assigned to one time instant, whereas a more instantaneous data is gathered in the latter. In any case, both systems showed very similar trends beyond puntual differences (Fig. 5c). This explanation has been included in the first paragraph of the *Smart facemask performance* section:

“The variation of the responses of both sensors can be explained in terms of the different gas sampling procedures. Reference system was set to pump gas inside the sensor chamber during 10 s before each measurement in order to provide stable readings. On the other hand, facemask sensor response lays on the absorption/desorption balance of CO₂ inside the sensing membrane at every time. Therefore, in the former an integrated measurement is obtained but assigned to one time instant, whereas a more instantaneous data is gathered in the latter. In any case, both systems showed very similar trends beyond puntual differences.”

- It would be useful to outlook in more details how the system could be expanded in terms of more measurements and physiological parameters.

Response: We agree with the reviewer that this prototype could be expanded in the future in order to increase the versatility and contribute to a much more complete health monitoring system. In the manuscript, Table 2 shows a comparison of sensing smart facemasks found in literature where it can be seen a variety of parameters of interest for this kind of application: ammonia, ethanol, humidity, etc. There already exists a wide diversity of developed optical chemical sensors for measuring gases concentration and other analytes that could be adapted to be used with this system, thus providing more information of the user’s health status. As stated before, the main advantage of optical chemical sensors is the possibility of miniaturization, and as it was shown in previous works of our group, they could be combined in one single system to measure different parameters due to the selective nature of these sensors. In the work *“Flexible Passive near Field*

*Communication Tag for Multigas Sensing*⁶ it is proved how it is possible to use up to four colour digital sensors using a single flexible passive NFC tag, fulfilling the low-power consumption requirement. Furthermore, among other potentialities, it would be possible to include a microfluidic system (μ CAD) in contact with the subject skin to include sweat pH monitoring⁷. This assessment has been briefly included in the last paragraph of the **Discussion** section:

*“All these features strengthen the potential applications of the proposed device in the fields of non-invasive health monitoring, preclinical research, **prognostics** and diagnostics with wearable electronics. Furthermore, the versatility of the system could be the object of future research since it is possible to combine several optical chemical sensors in a single flexible passive NFC tag³⁰ to determine several gases concentrations of interest for health status.”*

- Please check all numbering of figures as some typos have been found, e.g. at line 297-298 the text refers to the wrong figures in the supplementary information.

Response: We apologize for these typos and we want to thank the reviewer for pointing them out. We have gone through the whole text to check the numbering of all figures and correct minor typing mistakes.

● Reviewer #2

This manuscript depicted a smart mask sensing platform for gaseous CO₂ monitoring with near-field-enabled data and power transmission. Although wearable electronics, especially smart mask, got tremendous attention due to need for tele-healthcare, the CO₂ monitoring in this project is questionable. First, because the gas exchange of FFP2 facemasks with filtering facepiece capacity is very poor, they are only recommended for special work situations rather than daily wear. For daily protection during the epidemic, even medical staff only need to wear medical masks. Hence, the use of monitoring CO₂ in FFP2 masks seems not obvious for epidemic devices.

Response: We thank the reviewer for this valuable feedback. Regarding the FFP2 facemasks recommendation mentioned by the reviewer, we would like to highlight that the proposed paper is entirely limited to the development of a sensing platform for gaseous CO₂ determination inside such facemasks. The discussion whether FFP2 facemasks should be worn only for special situations or on a daily basis falls out of the scope of this work and is outside our area of expertise. In any case, the developed CO₂ sensing platform is just a system proposal for this type of facemasks, regardless of how or when they are used. However, it is worth noting that, according to very recent studies, SARS-CoV-2 is evolving toward more efficient aerosol generation and loose-fitting masks could provide only modest source control⁸. Therefore, it cannot be discarded that tight-fitting masks and respirators such as FFP2 masks would be recommended in more scenarios because of new virus variants. This comment has been briefly included at the end of the first paragraph of the **Introduction**:

*“This issue has raised great controversy regarding the actual **effectiveness** of facemasks on the disease control against their possible adverse effects⁴⁻⁶. It is not the aim of this report to directly tackle this controversy, but designing and evaluating a battery-free and wearable solution to monitor carbon dioxide (CO₂) concentration into the dead space volume (DSV) of a filtering facepiece FFP2 facemask. **However, it is worth noting that, according to very recent studies, SARS-CoV-2 is evolving towards more efficient aerosol generation and loose-fitting masks could provide only modest source control⁷”.***

Second concern is about “real-time monitoring”. The system needs to actively provide power through an external NFC chip of external equipment such as a smartphone. Otherwise, it can't transmit data in real time. It's not really “real-time monitoring”. it is just wireless data transmission.

Response: We thank the reviewer for this interesting observation. Even though we understand the source of concern, we humbly disagree with the comment related to the ‘real-time’ nature of the developed system. The reviewer is right that the system needs an external reader such as a smartphone to be powered and conduct a CO₂ measurement. However, this requirement does not conflict with the fact that the conducted measurement is in ‘real time’, since the system delivers the CO₂ gas concentration at the very moment of the measurement (therefore, in ‘real time’). In fact, a quick literature search delivers thousands of references where passive systems (i.e. battery-less RFID/NFC tags) are classified as real-time monitoring systems. Perhaps the reviewer refers to the fact that the monitoring is not continuous, which is true due to the absence of a battery for data logging when the reader is not powering the system. In any case, to avoid potential confusion for the readers, this non-continuous nature of the measurement system has been clarified in the first paragraph of the **General architecture of the system** section:

*“The system allows CO₂ **readout** by means of a deposited novel chemical sensor with fluorescent response, whose real-time results are **shown** in an NFC-enabled smartphone application **when it is approached to the tag (Fig. 1)**. [...] **In this way, a measurement of the CO₂ gas concentration is available virtually immediately (i.e. in real time) upon approach of the smartphone, although the monitoring is not continuous since it needs the powering from the smartphone acting as the NFC reader. In return, the***

system is battery-free, which has been possible by designing the electronics using ultra low power consumption components.”

Additionally, the word ‘monitoring’ has been replaced by the words ‘determination’/‘readout’ throughout the whole revised manuscript (including the title) to avoid any wrong assumptions about continuous monitoring.

Thirdly, the effect on gas permeability of mask itself needs to be assessed due to the final dimensions (60×45 mm²) of the device based on PET is it takes up a quarter of the mask area.

Response: A study of the influence of the PET passive tag on facemask gas permeability has been conducted. Measurements have been taken every 30 seconds during 20 minutes, with a FFP2 facemask with and without the tag placed inside, as it is shown in the manuscript (Fig. 1c). The equipment used for this experiment was the instrument employed as the reference instrument (MultiRAE Lite, RAE Systems, Inc. by Honeywell, CA, USA) for comparison with the developed system. The obtained results show that the measurement of CO₂ concentration in the presence of the PET tag is increased by 5% of relative CO₂ concentration both in static and during physical exercise. We believe that this slight increase can be comparable to that caused by facemask fitting variation among individuals. Nevertheless, the reduction of component size, as explained above, as well as the reduction of NFC antennas could lead to a decrease in the size of future tag designs. The size reduction will reduce this effect. This appreciation has been now included with the following sentence in the last paragraph of the *Design and performance of the sensing NFC flexible tag*:

“When placed on the inner face of a facemask as shown in Fig. 1b, the tag is completely unnoticeable to the user thanks to its reduced thickness and compact size, making it comfortable to wear within the facemask. Furthermore, the presence of the PET tag does not significantly affect the gas permeability of the facemask, only increasing the CO₂ concentration in the DSV by a relative 5% compared to the tagless facemask.”

Fourthly, in practical application, the red coordinate by photodetector is susceptible to the ambient light. The accuracy of sensors under different ambient light is uncertain.

Response: The reviewer is right that the red coordinate delivered by the digital photodetector can be susceptible to the ambient light. To quantify to what extent this occurs for practical applications, new experimentation has been conducted. To this end, the sensing tag has been placed inside an FFP2 facemask worn by a user under different ambient light conditions. In all cases, CO₂ concentration during the experiments was virtually constant at ~2.4%, which is a common value during low work-rate activities (see Fig. 5c-d). Each scenario has been quantified in terms of illuminance using a lux meter RS Pro LX-105 (RS Components, London, UK). To place the amount of lux into context, the table below shows some examples of the illuminance provided under various light conditions:

Environment	Typical illuminance (lux)
Night (no moon)	<0.01
Moonlight (full moon)	1
Public areas with dark surroundings	20-50
Office building hallway/toilet lighting	80

Train station platforms	150
School classroom/University lecture hall	250
Office/Laboratory/Kitchen	320-500
Sunrise and sunset	400
Factory/Supermarket	750
Overcast day	1k
Full daylight (no direct sun)	10k-25k
Direct sunlight	32k-100k

Two different colours of standard FFP2 facemasks have been tested: white and black. For each facemask colour, five different scenarios have been tested with a lux range from ~200 lx up to ~67 klx. To achieve the different light conditions up to ~3.5 klx, a dimmable LED ring light (ELEGANT EGL-02S) was employed, while for the two last conditions (>10 klx), the tests were conducted outdoors in the shadow and under direct sunlight, respectively. For each scenario, the average of three measurements of the R coordinate was taken and the relative error for each case was calculated from the measurements with and without illumination (in the dark, i.e. at 2 lx).

Based on the obtained results, a new graph has been prepared (Fig. 5c) and included in *Smart facemask calibration and characterization* section, showing the reading relative errors as a function of the luminous intensity. Three regions have been plotted as a visual guide: green zone for relative errors below 5%; orange for relative error values between 5% and 15%; and red zone for relative errors above 15%. The lines connecting the experimental data points are plotted only as eye guides. The graph shows the differences between the white and black facemasks in terms of measurement errors. As expected, black facemasks deliver lower error values since they achieve better light shielding than white facemasks. The vertical line in the graph differentiates between two regions, corresponding to indoors and outdoors light conditions. Predictably, the errors increase considerably outdoors, where only black face masks should be used to keep the errors below 8% even in direct sunlight. Indoors, measurement error is below 3% for all cases. Based on the obtained results, the following text has been added to the revised version of the manuscript:

- In the *Materials and Methods* section, the description of the conducted experiment has been included:

“Ambient light tests. Five different scenarios were tested with illuminance levels ranging from ~200 lx up to ~67 klx, which was quantified using a lux meter RS Pro LX-105 (RS Components, London, UK). To achieve the different light conditions up to ~3.5 klx, a dimmable LED ring light (ELEGANT EGL-02S) was employed, while for the two last conditions (>10 klx), the tests were conducted outdoors in the shadow and under direct sunlight, respectively. For each scenario, the average of three measurements of the R coordinate was taken and the relative error for each case was calculated from the measurements with and without illumination (in the dark, i.e. at 2 lx). Two different colours of standard FFP2 facemasks were tested: white and black. In all cases, CO₂ concentration during the experiments was virtually constant at ~2.4%, which is a common value during low work-rate activities.”

- In **Smart facemask calibration and characterization** section, the text has been updated to discuss the main results obtained from the new experimentation:

“Due to the nature of the sensing mechanism (i.e. based on optical measurements), environmental light can interfere with the system compromising the system performance. To assess the effect of ambient light on the system functionality, the sensing tag was placed inside an FFP2 facemask worn by the user under different ambient light conditions. Fig. 5c shows the measurement relative errors obtained as a function of the luminous intensity. Three regions have been plotted as a visual guide: green zone for relative errors below 5%; orange for relative error values between 5% and 15%; and red zone for relative errors above 15%. The vertical line in the graph differentiates between two regions, corresponding to indoor and outdoor light conditions. The graph shows the differences between the white and black facemasks with regard to optical shielding. As expected, black facemasks deliver lower error values since they achieve better light shielding for the optical measurements than white facemasks. Predictably, the errors increase considerably outdoors, where only black face masks should be used to keep the errors below 8% even in direct sunlight. In the case of indoor environments, also white facemasks provide enough light shielding to keep measurement errors below 3% for all cases, while black facemasks deliver errors below 1% in indoor scenarios.”

Figure 5. Smart facemask calibration and characterization. *a* Calibration curve and *b* temperature dependence of the CO_2 sensing membrane. *c* Ambient light influence: Reading relative error under different ambient light conditions (both indoors and outdoors) using the sensing system inside white and black FFP2 facemasks. The lines connecting the experimental data points are plotted as eye guides. *d* Sensor lifetime: Reading relative error over a continuous 12-hour period of time replicating the conditions inside a FFP2 facemask.

Finally, authors need to pay attention to the difference between LOD and sensitivity, as well highlight the value of the developed chip as a health monitoring system by pointing out whether the results can be used to prognosis the health status.

Response: We really appreciate this comment, since these concepts were introduced in a confusing way and some information was mistaken, indeed. The sensitivity is the smallest amount of change that our system is capable of providing for the measured parameter causing a read change equivalent to the resolution of the equipment. In the manuscript, we derived the value of resolution for CO₂ concentration using the experimental fitting function (Fig. 5b). This value differs from the limit of detection, which is the lowest analyte concentration detected when a blank sample is analysed. The LOD was calculated using the fluorescence spectrometer from the chemical sensor response and the standard criteria explained in section **CO₂ sensing scheme**. In the original version of the manuscript, there was a typo in Table 1, being the correct value of LOD 140 ppm, i.e. one order of magnitude lower. Now, Table 1 has been corrected, and the term LOD has been removed from section **Smart facemask calibration** to avoid misinterpretation:

“The resolution of the system can be derived from the fitting function³⁰. The value obtained in the worst case (at 4.8%) is 221 ppm, and in the range of interest the resolution achieved is 103 ppm, which is very close to the resolution provided by the manufacturer of the reference instrument described in the section below (100 ppm). Moreover, response and recovery times are below 1 s in this CO₂ concentration range.”

As mentioned by the reviewer, the developed tag is useful for health monitoring, but it is worth noting that it only provides information of CO₂ concentration at particular moments in time, and not in a continuous way. Therefore, it would be too ambitious to affirm the possibility of diagnosing the health status of a person just using these CO₂ measurements. Prognosis and diagnosis of the health status, regarding breathing diseases, implies many more physiological parameters such as breathing rate, oxygen saturation, tidal volume, etc. With respect to some other physical alterations, the proposed system could be useful for prognosis of headaches, dizziness, lightheadedness or fatigue if consecutive measurements of high levels of CO₂ in the dead space volume were maintained in time⁹⁻¹³. This appreciation has been clarified in the **Discussion** section:

*“All these abilities, combined with the wearable and wireless features of the system, make this powerful tool a strong candidate for adoption in numerous areas such as non-invasive monitoring of physical and sport activity, at workplace, or for preclinical research, **prognostics and diagnostics of illnesses** (e.g. in Chronic Obstructive Pulmonary Disease (COPD) patients) **together with other medical equipment**.”* and *“All these features strengthen the potential applications of the proposed device in the fields of non-invasive health monitoring, **preclinical research, prognostics and diagnostics** with wearable electronics.”*

Overall, this manuscript is limited in material innovation or advanced technological breakthrough, and thus is not recommended for publication in Nature Communications.

● Reviewer #3

Key results: The paper describes a new concept for CO₂ sensing in face masks. The system is built up completely with many different aspects of the engineering development (sensor, electronic hard- and software, communication) as well as physiological experiments being described. This completeness of the development approach makes the paper somehow unique. The CO₂ sensor and its measurements are tested in practical examples with good correlations demonstrating the practical usage potential of such system.

Response: We thank the reviewer for their positive comments and for their thoughtful and careful review towards improving our manuscript. Below are addressed all the comments point by point.

Validity, Significance: The data and descriptions taken for and from the system are well presented and the fact that the system practically works speak for itself in terms of the validity of the approach. Some statistical and long-term data i.e. for the sensor, it's cross-sensitivities and drifts are lacking, but only to an extend which goes beyond an initial prototype study (which the reviewer here considers not to be very relevant for the quality of the paper). The individual parts of the system described in the paper are not very well compared to the state of the art: while this is a difficult task and needs specifications to be benchmarked i.e. at the interfaces of the individual block/tasks described here, this would help readers to better understand significance and new aspect specific to this work. The authors lack from making clear what they do different here and why compared to other approaches for CO₂ sensing, for building NFC powered systems, and for their physiological studies (all of this could be put into respect to the application).

Data and methodology: The authors present enough data demonstrating the principle and its general suitability as a face-mask indicator for high CO₂ concentration there. It is on the one hand side a matter of the fact of the completeness of the problem they try to solve, that many aspects of the individual parts and its suitability in the system are missing, but on the other hand side it is not fully clear to the reviewer whether the experimental data taken here is sufficient to prove the feasibility of such approach in a broader context (e.g. different people, different micro- and macro-climate, ...). Therefore, in the 'suggested improvements' section a few suggestions are made.

Analytical approach, clarity and context: As mentioned above, a clear indication and discussion on the how and why certain approaches have been taken here is missing. This unfortunately reduces the impact of the paper with respect to education of readers and new ongoing research in this area. Some parts of the work are rather well explained (even beyond the level of newness) like the Android interface, however in other parts of the text more detail may be of higher scientific interest (see 'suggested improvements').

Response: We really appreciate these insightful comments, which have been very helpful in improving the manuscript. In line with your suggestions, the main novelties in the revised manuscript can be summarised in the following points:

- Details of and rationale for our design approaches and methodological choices, such as the election of NFC technology and the individual parts comprising the system.
- Discussion of why the CO₂ sensing mechanism was taken as the best approach in our case.
- Benchmarking of the realized CO₂ sensor with other CO₂ sensors that can be found in the literature.
- New experimentation to assess the effect of the ambient light on the system performance.
- New experimentation to assess gas permeability through the mask with the sensing system.
- New experimentation to analyse sensor reusability in terms of sensor operating life.
- Apart from the above, some more specific reviewer's 'suggested improvements' have been addressed and will be answered later on.

In essence, general design criteria have been based on those of Internet of Things (IoT) devices: *sensing, connectivity, integration, compactness, scalability, low energy consumption, and affordable cost*. These general criteria have been the basis for the selection of the specific component parts of the developed sensing system:

- *Sensing - CO₂ sensing mechanism*: A detailed rationale for the choice of the sensing mechanism has been added to *CO₂ sensing scheme* section:

“Existing methods for CO₂ sensing can be classified into four techniques: infrared absorptiometry, catalytic bead sensors, electrochemical devices, and optical sensors. Regarding infrared sensors, they are subject to more strong interference as well as an increase in the cost and the size of the final prototype due to the characteristics of available sensors. The catalytic bead sensors are based on flameless combustion, and the electrochemical sensors, although effective and reliable, can be cross-sensitive to other gases. Due to the requirement of low-power consumption of the proposed application, these techniques were discarded. Optical sensors (see a brief revision in Supplementary Table 1), based on the luminescence measurement of a very stable inorganic phosphorus modulated by an acid-base indicator depending on the CO₂ concentration, have been widely studied showing advantages that are really attractive for the developed system: possibility of miniaturization, electrical isolation and less sensitivity to noise.”

Moreover, we have included a State-of-the-Art brief revision in *Supplementary Section 1* (see Supplementary Table 1). A rationale for the sensor composition has been exposed as well:

“As a State-of-the-Art brief revision, Supplementary Table 1 summarizes the principal analytical characteristics of representative CO₂ luminescent gas sensors found in the literature. CO₂ detection chemistry for optical sensors is generally based on plastic solid-state sensor film that works with a colour-based or fluorescence-based pH indicator, such as α -naphtholphthalein or 8-hydroxypyrene-1,3,6-trisulfonate with pKa that matches CO₂ acidity. The limitation of this design arises from the poor stability of the membranes. Due to the small number of existing fluorescent indicators, a combination of luminescent dye with non-luminescent pH indicator has been used. This system works by both primary or secondary inner-filter effect or by resonance energy transfer (RET), measuring the intensity of the luminescence, the decay time, ratiometric signals or colour. The drawback of these strategies is the limited shelf life that arises from photobleaching of the dye, evaporation of its trace water content, and degradation of the quaternary amine hydroxide base. Different solutions have been tested, such as the inclusion of ionic liquids, or bases that do not suffer from Hoffman degradation. Another possible strategy in sensors based on secondary inner-filters is to replace the luminescent dye, whose emission is modulated by the pH indicator, by an LED with the same wavelength of the dye. Another approach developed to improve the stability of sensor membrane is to replace the luminescent dye, typically a metal complex or an organic molecule, by stable inorganic phosphors. This last solution has been adopted in this study, in which the luminescence emission of an inorganic phosphor is absorbed by the basic form of the pH indicator, thus increasing the luminescence when the CO₂ concentration increases.”

Supplementary Table 1. Comparison of luminescence-based sensors for CO₂ gas determination found in the literature.

Sensing chemistry	Technique	LOD (ppm)/ range (%)	Response time (s)	Precision (%)	Storage lifetime (months)	Ref.
HPTS/TBP/TOAOH	LI	0-1%	4.3	-	24-36 h	1
HPTS/TOAOH/sol gel/TiO₂	LI-A/DLR	80	50	-	7 days/in vacuum	2
Ru(dpp)/Sudan III/TOAOH	LI/LT	60	20-30	1.6	3/dark	3
PAACHl	FI	2	20-30	-	12/4-8°C	4
PtOEP/N/TOAOH	FI	0-40%	60	-	14 days/dark	5
NaYF₄:Yb/BTB/TBAO H	FI	1100	10	1%	-	6
GAB/aB/IL	I	2600	23	0.70	At least 120 days/ dark	7
Cl/ETH2412/TDDMA Cl	A	200	-	Cl	-	8
La₂O₃S:Eu/N/TMAOH/	C	140	9	4.41	140 days/dark	This work

HPTS: 8-hydroxypyrene-1,3,6-trisulfonate; TBP: Tributylphosphate; TOAOH: Tetraoctylammonium hydroxide; Ru-dpp): Ruthenium (II) tris (4,7-diphenyl-1,10-phenanthroline); PAACHl: Chlorophyllide b derivatized polyallylamine; PtOEP: Platinum octaethylporphyrin; N: α -Naphtholphthalein; BTB: Bromothymol blue; TBAOH: Tetrabutylammonium hydroxide; GAB: Cr(III)-doped gadolinium aluminium borate; IL: Ionic liquid; aB: AzaBodipy; Cl: Carbonate ionophore; TDDMACl: Tridodecylmethylammonium chloride; TMAOH: Tetramethylammonium hydroxide; LI: Luminescent intensity, A: Absorbance, R: Ratiometric, LT: Luminescent lifetime; DLR: Dual luminophore reference; C: Colour.

- **Connectivity - NFC technology:** We chose NFC because it fills some technological gaps that cannot be sufficiently addressed with other wireless technologies (such as other RFID bands, Bluetooth or Wi-Fi). In particular, its ability to simultaneously transfer data and, more importantly, power between devices offers the opportunity for the design of battery-free sensing systems. This means that NFC eliminates the requirement of including a battery in the tag, with all the benefits that it entails in terms of less environmental impact, lower weight, lower cost, less maintenance, and increased wearability and comfort. Additionally, the choice of NFC technology enables the use of any NFC-enabled smartphone as the remote reader instead of an ad-hoc reader, making this technology within the reach of virtually any user. Another advantage of NFC as compared to other technologies relies on the possibility to produce NFC tags by established high-volume manufacturing methods (for example, roll-to-roll) on flexible substrates such as polyethylene terephthalate (PET) or paper. Some of these advantages of NFC over other technologies were already mentioned in the first paragraph of the **Discussion** section, while others have been now included to clearly justify why NFC was chosen:

“As can be seen in Table 2, some other wireless solutions have been implemented in previous smart facemasks designs, such as UHF Radio Frequency Identification (RFID) or Bluetooth. In our case, NFC was chosen over Bluetooth or other wireless protocols because it has a key advantage: its ability to simultaneously transfer data and, more importantly, power between devices offers the

opportunity of battery-less communication, as the required energy to supply the system can be obtained from the electromagnetic field generated by the remote reader. This means that NFC eliminates the requirement of including a battery in the tag, with all the benefits that it entails in terms of less environmental impact, lower weight, lower cost, less maintenance, and increased wearability and comfort. Additionally, the choice of NFC technology over other protocols such as UHF RFID allows the use of any NFC-enabled smartphone as the remote reader instead of an ad-hoc reader, making this technology within the reach of virtually any user and thus enabling self-health management. Another advantage of NFC as compared to other technologies relies on the possibility to produce NFC tags by established high-volume manufacturing methods (for example, roll-to-roll) on flexible substrates such as PET or paper.”

- **Integration - PET substrate:** To integrate the system into the facemask, PET substrate was chosen because of its excellent combination of properties such as mechanical, thermal, chemical resistance and dimensional stability. In particular, the following three properties were key factors for our development: *i) It has a high optical transmittance of ~90%, which is needed to avoid the attenuation of the optical sensor signal; ii) It is highly flexible, allowing bending angles beyond those needed for the given application; and iii) It has excellent adherence and a high melting point (~270 °C), which are crucial characteristics for the successful deposition and thermal curing of the conductive Ag-based ink. All this information has been now included in the first paragraph of the **Design and performance of the sensing NFC flexible tag** section:*

“The designed tag was printed on 125 μm-thick PET flexible substrate. PET film was chosen because of the combination of three key properties required for our development: i) Its high optical transmittance of ~90%, needed to avoid the attenuation of the optical sensor signal; ii) Its high flexibility, which allows bending angles beyond those needed for the given application; and iii) Its excellent adherence and high melting point (~270 °C), which are crucial characteristics for the successful deposition and thermal curing of the conductive Ag-based ink.”

- **Compactness - Tag size:** The final tag size (60×45 mm²) was basically imposed by the design of the planar coil, whose details are given in the **Flexible tag fabrication and characterization** section (in **Materials and Methods**) and **Supplementary Section NFC antenna initial calculations and design**. Additionally to those explanations, the size choice was a compromise between achieving enough inductive coupling (which was assessed through the inductive coupling factor *k*, see Fig. 4e), while making the system as compact as possible by placing the circuitry on the inner space of the planar coil. This justification has been included in the first paragraph of the **Design and performance of the sensing NFC flexible tag** section:

“Photographs of the fabricated flexible tag are shown in Fig 3c-e, with final dimensions of 60×45 mm². Tag size was designed as a result of the compromise between achieving enough inductive coupling while making the system as compact as possible by placing the circuitry within the inner space of the planar coil.”

- **Low energy consumption - Low-power electronics design and component selection:** Built on the premise that a passive system was desired (i.e. without any battery), the electronic design was optimized for low-energy operation to allow power supply entirely by the EM field generated by the NFC-enabled smartphone. In this context, the search and selection of the electronic components was a key step for the battery-less operation of the system. Due to the limitations in terms of length, the discussion on the selection criteria of the different components has been mainly written in the **Supplementary Information**. A sentence has been added in the first paragraph of the **Design and performance of the sensing NFC flexible tag** section to clarify this:

“Fig. 3a shows the circuit block diagram of the system and its interconnections. The conditioning circuitry comprises as main electronic components *an extreme* low-power microcontroller unit (XLP MCU) for control and data processing purposes; an NFC Dynamic Tag IC along with a custom-designed NFC antenna for NFC link and power management; and a sensing module consisting of an UV LED, a digital colour sensor and a temperature sensor. *Additional details on the selection criteria of every component based on the system requirements can be found in Supplementary Section 3.*”

- **NFC chip:** The power consumption of the system was 4.45 mW in operation mode. Even if this can be considered as low-power consumption, not all the available NFC ICs in the market are able to provide that amount of energy at the required regulated voltage level and current³⁻⁵. The employed NFC chip was chosen because its energy harvesting capabilities allow a regulated supply power up to 22.5 mW (up to 5 mA @4.5V). This information has been now added to **Design and performance of the sensing NFC flexible tag** section:

“In this application, the power consumption of the whole system is 4.45 mW in operation mode, which is roughly 20% of the total available power that can be sourced from the chosen NFC chip, i.e. 22.5 mW. Low-power electronic design and component selection has made this possible, thus enabling battery-free operation of the system.”

As an example, there are some other solutions combining NFC chip and microcontroller unit in the same IC (e.g. RF430FRL15xH), but they are not able to provide the required power for the given application. Please refer to the answer to the next comment for more details about this matter.

- **Microcontroller Unit (MCU):** In line with the above, the MCU was chosen due to its eXtreme Low-Power (XLP family) features, low operating voltage range (1.8V to 3.6V) and digital/analog peripherals: serial communication (I²C and SPI), and 10-Bit Analog-to-Digital Converter (ADC). In fact, the MCU usage has been optimized in a way that all its available pins are used (see Fig. 3a). These criteria have been now included in the first paragraph of **Supplementary Section Sensing NFC tag components**:

“The MCU model was chosen due to its eXtreme Low-Power (XLP family) features, low operating voltage range (1.8 V to 3.6 V) and digital/analog peripherals: serial communication (I²C and SPI), and 10-Bit Analog-to-Digital Converter (ADC). All those features were required to communicate with the NFC chip through the SPI port; with the digital colour sensor through the I²C port; and to get the temperature readings by means of the ADC module. In fact, the MCU usage has been optimized in a way that all its available pins are used (see Fig. 3a).”

- **Digital colour detector:** This part was chosen because of a combination of several reasons: low voltage operation (2.5 V or 3.3 V), which was compatible with our requirements; low current consumption (75 μ A typical); it supports the I²C interface, which was necessary to communicate with the MCU; and its sensitivity to red emission (centered at $\lambda=615$ nm) overlaps the luminescent emission spectrum of La₂O₂S:Eu for the sensing mechanism (see Fig. 2b). The information in the first paragraph of **Supplementary Section Sensing NFC tag components** has been now expanded to include more details:

“The digital colour sensor used in this design is the S11059-02DT (Hamamatsu Photonics, Hamamatsu, Japan), which is an I²C interface-compatible digital detector sensitive to red ($\lambda_{peak} = 615$ nm), green ($\lambda_{peak} = 530$ nm), blue ($\lambda_{peak} = 460$ nm), and near infrared ($\lambda_{peak} = 855$ nm) incident radiation, which is codified in the corresponding four 16-bit digital words. Its sensitivity

to red emission (centered at $\lambda=615$ nm) overlaps the luminescent emission spectrum of $\text{La}_2\text{O}_2\text{S}:\text{Eu}$ for the sensing mechanism (see Fig. 2b). Besides its high resolution, this detector was selected due to its low voltage operation (2.5 V or 3.3 V) and low power consumption of only 250 μW in operation mode, which makes it optimal for our battery-less design.”

- **LED:** The exciting LED was selected on the basis of the absorption spectrum of the $\text{La}_2\text{O}_2\text{S}:\text{Eu}$ phosphor, which is shown in Fig. 2b. It can be seen that the maximum peak occurs at 325 nm, so ideally the LED should have its emission spectra at this wavelength to optimize the sensing mechanism. Unfortunately, we have not been able to find any commercially available LED at that particular frequency that, additionally, meets our design requirements regarding low polarization voltage, low-power consumption and small size. The chosen LED model that meets such requirements has its emission spectra at 367 nm, which is not the optimum value but still overlaps a valid band of the absorption spectrum of the $\text{La}_2\text{O}_2\text{S}:\text{Eu}$ phosphor, as can be seen in Fig. 2b. This explanation has been now included in *Supplementary Section Sensing NFC tag components*:

“The UV LED was selected on the basis of the absorption spectrum of the $\text{La}_2\text{O}_2\text{S}:\text{Eu}$ phosphor, in which the maximum peak occurs at 325 nm (see Fig. 2b). Therefore, the LED was selected to have its peak emission as close as possible to this wavelength to optically excite the CO_2 sensitive membrane, whilst meeting our design requirements in terms of low polarization voltage, low-power consumption and small size. To the best of the authors’ knowledge, the selected LED model with peak emission at 367 nm was the optimum one meeting the design requirements as far as possible. Once excited by the UV LED, the luminescent response delivered by the CO_2 sensitive membrane is detected by the digital colour sensor and sent to the MCU for further processing.”

- **Temperature sensor:** The temperature sensor model MCP9700A was used to correct the temperature drifts of the chemical sensor. This model was chosen because of its miniature size ($2.90 \times 1.30 \times 0.95$ mm³) and because it presents an accuracy of ± 1 °C and a very low current consumption of only 6 μA . This justification is now included in *Supplementary Section Sensing NFC tag components*:

“A temperature sensor model MCP9700A (Microchip Technology Inc.) was also included in the design, connected to one 10-bit Analog to Digital Converter (ADC) input of the MCU, to correct the temperature drifts of the chemical sensor. This sensor was chosen due to its miniature size ($2.90 \times 1.30 \times 0.95$ mm³) and because it presents an accuracy of ± 1 °C and a very low current consumption of only 6 μA .”

- **Affordable cost:** The cost of the developed sensing system has been estimated considering the cost of its different components, where the ICs are the most expensive ones. In this way, we estimate a large-scale production cost below 5 € for the whole sensing system. This information has been included in the first paragraph of *Design and performance of the sensing NFC flexible tag* section of the revised manuscript:

“Considering the cost of the different parts comprising the sensing tag, where the ICs are the most expensive components, the estimated large-scale production cost of the whole sensing system is below 5 €. As for the sensing membrane, the cost can be estimated below 0.01 € without considering mass production.”

Suggested improvements: Below you can find a list of suggestions and items to improve with priority according to the listing:

- General structure: the NFC system as shown here and its electrical behavior is very close to industry standards and commercial hardware in many aspects. The parts used here can be bought commercially even at higher integration level for such sensors (e.g. Texas Instruments RF430FR215x). However, the integration on flexible substrates and its consequences for the circuits (e.g. may cause strain changes at the device level with impact on electrical behavior) as well as the approach to measure and package the sensor are really new and are only described here on a very low level of detail (see later suggestions):

Response: The reviewer has rightly mentioned a very interesting component by Texas Instruments, the NFC transponder chip RF430FRL15xH. Indeed, the most attractive feature of this NFC chip is that it embeds a programmable low-power microcontroller unit (MCU), so in principle it could combine our two separate components (NFC chip + MCU) into just one chip. In fact, this IC was one of the candidates during the design phase of the system, since the authors already had previous experience with it¹⁴. However, the design requirements of the smart facemask system in terms of power consumption were not met by the energy harvesting capabilities of this chip. The technical documentation shows¹⁵ that the maximum power that can be sourced from the RF430FRL15xH is limited to just 150 μ A at 2.9 V (i.e., less than 0.5 mW), which is not enough in our case to supply the circuitry, even though the design was optimized for very low power consumption. That is the reason why we had to utilize different components for the NFC chip and the MCU. In particular, the energy harvesting capabilities of the chosen NFC chip allow a regulated supply power up to 22.5 mW, being the power consumption of our system of 4.45 mW in operation mode.

Regarding the system integration on flexible substrate and its consequences for the circuit at electrical level, the conducted study upon bending curvatures up to $\sim 110^\circ$ are beyond the maximum bending angles that facemasks go through under normal and proper use (estimated at $\sim 20^\circ$ - 30°). It is worth pointing out that the conducted analysis of the reading distance as a function of the bending angle (see Fig. 4f) also involves the correct operation and functionality of the whole sensing system in such scenarios. Therefore, strain changes falling under the studied bending ranges only affect the system in terms of reading distance. Nevertheless, we recognize that the physical integrity of the system might be affected by incorrect and improper use of the facemask in terms of excessive bending, twisting or crunching, so this limitation has been mentioned in the paper by adding the following sentence in the last paragraph of *Design and performance of the sensing NFC flexible tag* section:

“Further, since the developed system is flexible, different curvatures were tested to study their impact on the tag’s operation and reading distance. As shown in Fig. 4f, the sensing system operates correctly with a reading distance above 10 mm up to a bending angle of $\sim 110^\circ$ along both bending axes. This bending value is beyond the required ones for the given application, since the tag inside the facemask lies virtually flat. However, there might be occasions where some bending is needed depending on the shape of the user face. Even so, we estimate maximum bending angles in practice between $\sim 20^\circ$ - 30° . Therefore, the tag would still be readable with sufficient range for the given application in all cases. Nevertheless, it is worth mentioning that due to the attachment of rigid ICs to the flexible substrate, the physical integrity of the system might be affected by incorrect and improper use of the facemask in terms of excessive bending, twisting or crunching.”

Related to the approach to measure and package the sensor, the following text has been included in Supplementary Section 1:

“Regarding the approach to measure and package the sensor, the strategy used is based on the displacement of the acid-base equilibrium of a pH indicator immobilized on the membrane in the presence of CO₂. To increase the sensitivity of the sensor towards CO₂, an inert luminescent dye, whose emission band overlaps the absorption band of the pH indicator in its basic form, is co-immobilized on the membrane. In absence of CO₂, the entire indicator will be in basic form due to the presence of a quaternary ammonium hydroxide. The luminescence of the dye will be completely cancelled by a secondary internal filter mechanism. In the presence of CO₂, there will be a shift towards the acid form of the indicator with the consequent increase in luminescence. A stable inorganic phosphor is selected as luminescent material, which increases the stability and lifetime of the membrane. Additionally, an ionic liquid is included in the membrane composition to strengthen the sensor stability and sensitivity by increasing the solubility of CO₂ in the membrane. In addition, a surfactant is included to reduce the interfacial tension and facilitate the permeation of CO₂ gas in the membrane. All these components are encapsulated in a hydrophilic hydroxypropylmethylcellulose membrane to prevent water loss over the life of the membrane.”

- o It is not clear to the reviewer whether the NFC powering and exposure to a magnetic field causes side effects to the CO₂ measurement accuracy – and if not: how this is avoided?

Response: Spin chemistry teaches us that magnetic fields can influence chemical reactions that involve radical intermediates altering the rate, yield, or product distribution¹⁶. In this case, the chemical reactions used for CO₂ recognition are based on the modification of the acid base equilibrium position of α -naphtholphthalein and do not involve radical intermediates. In any case, according to COMSOL simulations, the NFC induced magnetic field in the tag falls into very low values, between 25 and 10 μ T, without expected effect neither on the chemical sensor, nor on the electronic components. This information has been included in the last paragraph of *Supplementary Section NFC antenna characterization*.

“Regarding the exposure of the CO₂ sensor to the magnetic field induced by the NFC reader, spin chemistry teaches us that magnetic fields can influence chemical reactions that involve radical intermediates altering the rate, yield, or product distribution²⁰. In this case, the chemical reactions used for CO₂ recognition are based on the modification of the acid base equilibrium position of α -naphtholphthalein and do not involve radical intermediates. In any case, according to COMSOL simulations, the NFC induced magnetic field in the tag falls into very low values, between 25 and 10 μ T, without expected effect neither on the chemical sensor, nor on the electronic components.”

- o The same question arises with stray light – here an answer is given, but the reviewer would also be interested in the effects of air humidity in this system and its correlation the CO₂ measurement principle

Response: The reviewer is right that the red coordinate delivered by the digital photodetector can be susceptible to the ambient light. To quantify to what extent this occurs for practical applications, new experimentation has been conducted. To this end, the sensing tag has been placed inside an FFP2 facemask worn by a user under different ambient light conditions. In all cases, CO₂ concentration during the experiments was virtually constant at \sim 2.4%, which is a common value during low work-rate activities (see Fig. 5c-d). Each scenario has been quantified in terms of illuminance using a lux meter RS Pro LX-105 (RS Components, London, UK). To

place the amount of lux into context, the table below shows some examples of the illuminance provided under various light conditions:

Environment	Typical illuminance (lux)
Night (no moon)	<0.01
Moonlight (full moon)	1
Public areas with dark surroundings	20-50
Office building hallway/toilet lighting	80
Train station platforms	150
School classroom/University lecture hall	250
Office/Laboratory/Kitchen	320-500
Sunrise and sunset	400
Factory/Supermarket	750
Overcast day	1k
Full daylight (no direct sun)	10k-25k
Direct sunlight	32k-100k

Two different colours of standard FFP2 facemasks have been tested: white and black. For each facemask colour, five different scenarios have been tested with a lux range from ~200 lx up to ~67 klx. To achieve the different light conditions up to ~3.5 klx, a dimmable LED ring light (ELEGANT EGL-02S) was employed, while for the two last conditions (>10 klx), the tests were conducted outdoors in the shadow and under direct sunlight, respectively. For each scenario, the average of three measurements of the R coordinate was taken and the relative error for each case was calculated from the measurements with and without illumination (in the dark, i.e. at 2 lx).

Based on the obtained results, a new graph has been prepared (Fig. 5c) and included in ***Smart facemask calibration and characterization*** section, showing the reading relative errors as a function of the luminous intensity. Three regions have been plotted as a visual guide: green zone for relative errors below 5%; orange for relative error values between 5% and 15%; and red zone for relative errors above 15%. The lines connecting the experimental data points are plotted only as eye guides. The graph shows the differences between the white and black facemasks in terms of measurement errors. As expected, black facemasks deliver lower error values since they achieve better light shielding than white facemasks. The vertical line in the graph differentiates between two regions, corresponding to indoors and outdoors light conditions. Predictably, the errors increase considerably outdoors, where only black face masks should be used to keep the errors below 8% even in direct sunlight. Indoors, measurement error is below 3% for all cases. Based on the obtained results, the following text has been added to the revised version of the manuscript:

- In the **Materials and Methods** section, the description of the conducted experiment has been included:

“Ambient light tests. Five different scenarios were tested with illuminance levels ranging from ~200 lx up to ~67 klx, which was quantified using a lux meter RS Pro LX-105 (RS Components, London, UK). To achieve the different light conditions up to ~3.5 klx, a dimmable LED ring light (ELEGANT EGL-02S) was employed, while for the two last conditions (>10 klx), the tests were conducted outdoors in the shadow and under direct sunlight, respectively. For each scenario, the average of three measurements of the R coordinate was taken and the relative error for each case was calculated from the measurements with and without illumination (in the dark, i.e. at 2 lx). Two different colours of standard FFP2 facemasks were tested: white and black. In all cases, CO₂ concentration during the experiments was virtually constant at ~2.4%, which is a common value during low work-rate activities.”

- In **Smart facemask calibration and characterization** section, the text has been updated to discuss the main results obtained from the new experimentation:

Figure 5. Smart facemask calibration and characterization. *a* Calibration curve and *b* temperature dependence of the CO₂ sensing membrane. *c* Ambient light influence: Reading relative error under different ambient light conditions (both indoors and outdoors) using the sensing system inside white and black FFP2 facemasks. The lines connecting the experimental data points are plotted as eye guides. *d* Sensor lifetime: Reading relative error over a continuous 12-hour period of time replicating the conditions inside a FFP2 facemask.

“Due to the nature of the sensing mechanism (i.e. based on optical measurements), environmental light can interfere with the system compromising the system performance. To assess the effect of ambient light on the system functionality, the sensing tag was placed inside

an FFP2 facemask worn by the user under different ambient light conditions. Fig. 5c shows the measurement relative errors obtained as a function of the luminous intensity. Three regions have been plotted as a visual guide: green zone for relative errors below 5%; orange for relative error values between 5% and 15%; and red zone for relative errors above 15%. The vertical line in the graph differentiates between two regions, corresponding to indoor and outdoor light conditions. The graph shows the differences between the white and black facemasks with regard to optical shielding. As expected, black facemasks deliver lower error values since they achieve better light shielding for the optical measurements than white facemasks. Predictably, the errors increase considerably outdoors, where only black face masks should be used to keep the errors below 8% even in direct sunlight. In the case of indoor environments, also white facemasks provide enough light shielding to keep measurement errors below 3% for all cases, while black facemasks deliver errors below 1% in indoor scenarios.”

In the case of air humidity, the dependency of the R colour coordinate of the CO₂ sensing membrane luminescence with relative humidity (RH) has been investigated using a thermostatic chamber. This has been included in last paragraph of **Smart facemask calibration** section:

“Besides, a humidity sweep study was conducted at 30 °C in the range from 10% to 90% RH. The obtained results indicate that the R coordinate of the sensing membrane is independent of the humidity in the ambient air. This is a very remarkable result since it eliminates the need for monitoring RH inside the facemask, which is around 70% on average under normal circumstances.”

Putting less emphasis on the standard parts of the system and more on how to optimize for the purpose would most likely be appreciated by most readers.

Response: We hope that the detailed answers given to the previous comments can serve to justify the different design and selection criteria as well as the methodological approaches that we have taken to optimize for the purpose of this work.

- In a similar context, more details about how the wavelength planning for good sensitivity in this system, the optimization of the solution and the packaging of the sensor material are done would be appreciated as well...

Response: As mentioned above, the exciting LED was selected on the basis of the absorption spectrum of the La₂O₂S:Eu phosphor, which is shown in Fig. 2b. It can be seen that the maximum peak occurs at 325 nm, so ideally the LED should have its emission spectrum at this wavelength to optimize the sensing mechanism. The absorption spectrum of the phosphor is settled by its chemical composition and it cannot be changed. However, on the other hand, this spectrum is broad enough to consider other choices. Thus, the chosen LED model, which meets the requirements of low polarization voltage, low-power consumption and small size, has an emission spectra at 367 nm, which is not the optimum value but still overlaps a valid band of the absorption spectrum of the La₂O₂S:Eu phosphor, as can be seen in Fig. 2b. This explanation has been now included in **Supplementary Section Sensing NFC tag components:**

“The UV LED was selected on the basis of the absorption spectrum of the La₂O₂S:Eu phosphor, in which the maximum peak occurs at 325 nm (see Fig. 2b). Therefore, the LED was selected to have its peak emission as close as possible to this wavelength to optically excite the CO₂ sensitive membrane, whilst meeting our design requirements in terms of low polarization voltage, low-power consumption

and small size. To the best of the authors' knowledge, the selected LED model with peak emission at 367 nm was the optimum one meeting the design requirements as far as possible. Once excited by the UV LED, the luminescent response delivered by the CO₂ sensitive membrane is detected by the digital colour sensor and sent to the MCU for further processing."

Related to the solution of the packaging of the sensor material, the following text has been included in Supplementary Section 1:

"Regarding the approach to measure and package the sensor, the strategy used is based on the displacement of the acid-base equilibrium of a pH indicator immobilized on the membrane in the presence of CO₂. To increase the sensitivity of the sensor towards CO₂, an inert luminescent dye, whose emission band overlaps the absorption band of the pH indicator in its basic form, is co-immobilized on the membrane. In absence of CO₂, the entire indicator will be in basic form due to the presence of a quaternary ammonium hydroxide. The luminescence of the dye will be completely cancelled by a secondary internal filter mechanism. In the presence of CO₂, there will be a shift towards the acid form of the indicator with the consequent increase in luminescence. A stable inorganic phosphor is selected as luminescent material, which increases the stability and lifetime of the membrane. Additionally, an ionic liquid is included in the membrane composition to strengthen the sensor stability and sensitivity by increasing the solubility of CO₂ in the membrane. In addition, a surfactant is included to reduce the interfacial tension and facilitate the permeation of CO₂ gas in the membrane. All these components are encapsulated in a hydrophilic hydroxypropylmethylcellulose membrane to prevent water loss over the life of the membrane."

- The measurement of the CO₂ takes rather long (9s – as stated in table 1). For an NFC powered system using a cell phone as a power source next to it this appears to be rather long to the reviewer. The reviewer however assumes the measurement does not need all power of the system all time. Why not using a small battery back-up via a super-capacitor or thin film battery to enhance the measurement time with respect to the read time? (e.g. the Texas Instrument chip mentioned above e.g. has power management support capabilities for such systems and this concept is used in commercial sensor system like blood sugar measurement (Abbott))?

Response: As the reviewer states, a response time of 9 s is indicated in Table 1. However, this time refers to the chemical sensing membrane, which takes 9 s to respond to CO₂ changes when the concentration is increased from 10% to 90% (i.e. 9 s is the rise time). In this particular facemask application, the range of interest for CO₂ varies from 0% to 5% and, within this interval, the sensing membrane takes less than a second for reacting to variations as it is stated in section **Smart facemask calibration**:

"Moreover, response and recovery times are below 1 s in this CO₂ concentration range."

Therefore, the CO₂ measurement using the NFC link is virtually immediate once the smartphone approaches the FFP2 facemask, and the power provided is enough for this purpose. This can be seen in the *Supplementary Movie* from minute 1:20 onwards. In addition, the use of batteries, even a thin one, involves disadvantages such as discomfort and less wearability, higher environmental impact, and an increase in weight, size and cost of the prototype.

- The android app programmed here is needed for the system demonstration, however the general structure of the firmware is rather straightforward, while corrections for drifts, sensor calibration, or time delay are not detailed well. This (or maybe the authors know even better examples of novelty in the context of this application) however, is what the reader of this Journal wants to know!?

Response: To further detail how the CO₂ value is computed within the firmware of the developed Android app (considering temperature drift correction and sensor calibration), the following explanation has been included in *Smartphone application* section (now in the SI document):

“When the user clicks on the Get Value button through the application user interface, such locations of the EEPROM memory are accessed by the application through the ISO14443A NFC protocol. Firstly, using the obtained temperature value, the measured R coordinate is corrected to compensate for the temperature drift according to the experimental temperature dependence (see Fig. 5b). Then, the CO₂ concentration is computed from the corrected R coordinate as per the obtained calibration curve (see Fig. 5a). Finally, all the information is displayed on the smartphone screen.”

In addition, we have updated the flowchart of the programmed application (*Supplementary Fig. 4*) to include such details. Therefore, the blue box which previously was simply labelled as “Compute CO₂ value” has been split into smaller subtasks to better explain the conducted procedure:

Supplementary Figure 4. Flowchart of the custom-developed smartphone application.

- Figure 5:

- o In subfigure c, the signal from the face mask sensor seems to somehow integrate the other sensors CO₂ measurement values – is this correct, or are these statistical outliers?

Response: The different gas sampling methods carried out by both measurement systems can explain these obtained signals. Reference system was configured to pump gas inside the sensor chamber for 10 s before stable readings can be obtained. On the other hand, facemask sensor response lays on the absorption/desorption balance of CO₂ inside the sensing membrane at the time. Therefore, in the former an integrated measurement is obtained but assigned to one time instant, whereas a more instantaneous data is gathered in the latter. In any case, both systems showed very similar trends beyond puntual differences (Fig. 5c). This explanation has been included in the first paragraph of the *Smart facemask performance* section:

“The variation of the responses of both sensors can be explained in terms of the different gas sampling procedures. Reference system was set to pump gas inside the sensor chamber during 10 s before each measurement in order to provide stable readings. On the other hand, facemask sensor response lays on the absorption/desorption balance of CO₂ inside the sensing membrane at every time. Therefore, in the former an integrated measurement is obtained but assigned to one time instant, whereas a more instantaneous data is gathered in the latter. In any case, both systems showed very similar trends beyond puntual differences.”

- o It is maybe beyond the expertise of the authors (and it is definitely beyond the expertise of the reviewer), but the time delayed correlation between cycling power and CO₂ as well as heart rate seems to be an interesting result which may trigger further investigation and also could lead toward new research aspects?

Response: Many thanks for your precise observation. In fact, we estimated this delay in our experiment, whose discussion is beyond our expertise. Because of your comment, we have rewritten that sentence and included this datum in the manuscript in the last paragraph of *Smart facemask performance* section:

“Interestingly, this experiment also revealed a time-delayed correlation between the cycling power and the measured CO₂ concentration by the smart facemask, as well as with the HR during the exercise, as observed from Fig. 5e-f⁵. In fact, we have estimated that the HR response was delayed 3 min with respect to CO₂ variations during this experiment. Although this interesting result is beyond the scope of our paper, it may trigger further investigations and research in this field.”

- Two minor issues:

- o Line 328: What does HR mean here? – later it is defined as ‘Heart Rate’ (line 374), but this does not make sense here!?

Response: We apologize for this mistake, which has understandably caused confusion to the reviewer. In line 328, it should appear “RH” instead of “HR” since we refer to “Relative Humidity” (RH), and not “Heart Rate” (HR). We also forgot to define the RH acronym prior to its use, so the revised text has now been updated to correct this mistake:

“The dependencies of the R colour coordinate of the CO₂ sensing membrane luminescence with temperature and relative humidity (RH) were also monitored using a thermostatic chamber VCL4006 (Vötsch Industrietechnik, Germany).

- o Line 93: typo ‘showed’ -> ‘shown’

Response: Many thanks for pointing out this typo, we have corrected it. We have also gone through the whole text to check and correct any other typing mistakes.

References

1. Heo, S. Y. *et al.* Wireless, battery-free, flexible, miniaturized dosimeters monitor exposure to solar radiation and to light for phototherapy. *Science Translational Medicine* **10**, eaau1643 (2018).
2. *Flexible Hybrid Electronics 2020-2030: Applications, Challenges, Innovations and Forecasts.* (2020).
3. Cao, Z. *et al.* Near-Field Communication Sensors. *Sensors* **19**, 3947–3947 (2019).
4. Kang, S.-G., Song, M.-S., Kim, J.-W., Lee, J. W. & Kim, J. Near-Field Communication in Biomedical Applications. *Sensors* **21**, 703–703 (2021).
5. Lazaro, A., Villarino, R. & Girbau, D. A Survey of NFC Sensors Based on Energy Harvesting for IoT Applications. *Sensors* **18**, 3746–3746 (2018).
6. Escobedo, P. *et al.* Flexible Passive near Field Communication Tag for Multigas Sensing. *Analytical Chemistry* **89**, 1697–1703 (2017).
7. Escobedo, P. *et al.* Wireless wearable wristband for continuous sweat pH monitoring. *Sensors and Actuators B: Chemical* **327**, 128948 (2021).
8. Adenaiye, O. O. *et al.* Infectious SARS-CoV-2 in Exhaled Aerosols and Efficacy of Masks During Early Mild Infection. *Clinical Infectious Diseases* (2021) doi:10.1093/cid/ciab797.
9. Li, Y. *et al.* Effects of wearing N95 and surgical facemasks on heart rate, thermal stress and subjective sensations. *International Archives of Occupational and Environmental Health* **78**, 501–509 (2005).
10. Beder, A., Büyükkoçak, Ü., Sabuncuoğlu, H., Keskil, Z. A. & Keskil, S. Preliminary report on surgical mask induced deoxygenation during major surgery. *Neurocirugía* **19**, 121–126 (2008).
11. Epstein, D. *et al.* Return to training in the COVID-19 era: The physiological effects of face masks during exercise. *Scandinavian Journal of Medicine & Science in Sports* **31**, 70–75 (2021).
12. Liu, C., Li, G., He, Y., Zhang, Z. & Ding, Y. Effects of wearing masks on human health and comfort during the COVID-19 pandemic. *IOP Conference Series: Earth and Environmental Science* **531**, 012034–012034 (2020).
13. Fikenzer, S. *et al.* Effects of surgical and FFP2/N95 face masks on cardiopulmonary exercise capacity. *Clinical Research in Cardiology* **109**, 1522–1530 (2020).
14. Escobedo, P., Bhattacharjee, M., Nikbakhtnasrabadi, F. & Dahiya, R. Smart Bandage With Wireless Strain and Temperature Sensors and Batteryless NFC Tag. *IEEE Internet of Things Journal* **8**, 5093–5100 (2021).
15. Kozitsky, A. & Jacobi, R. Frequently asked questions for RF430FRL15xH devices. (2019).
16. Rodgers, C. T. Magnetic field effects in chemical systems. *Pure and Applied Chemistry* **81**, 19–43 (2009).

Reviewers' Comments:

Reviewer #1:

Remarks to the Author:

The manuscript "Smart facemask for wireless CO₂ optical determination through flexible battery-less NFC tag" has been extensively revised and most of the questions have been addressed.

Minor revisions are still needed. In particular:

- The sentence "While standard facemasks simply act as air filters for the nasal and/or mouth passage, the integration of sensors to control parameters of interest can be an added value to improve their usage and effectiveness, including the developed smart facemask into the wearable devices of the Internet of Things (IoT) paradigm." is still confusing. I would suggest to remove "including the developed smart facemask into the wearable devices of the Internet of Things (IoT) paradigm."
- A full justification of the cost estimate should be included in the SI.

Reviewer #2:

Remarks to the Author:

The authors made significant improvement of the manuscript based on the comments and it is ready for publication.

Reviewer #3:

Remarks to the Author:

The list of suggestions and items to improve from the prior version are all revised in the new version.

Response to the Comments of the Reviewers

Title (previous): Smart facemask for wireless CO₂ optical determination through flexible battery-less NFC tag

Title (new): Smart facemask for wireless CO₂ monitoring

Authors: P. Escobedo, M.D. Fernández-Ramos, N. López-Ruiz, O. Moyano-Rodríguez, A. Martínez Olmos, I.M. Pérez de Vargas-Sansalvador, M.A. Carvajal, L.F. Capitán-Vallvey, A.J. Palma

Reference: NCOMMS-21-30539A

● **Reviewer #1**

The manuscript "Smart facemask for wireless CO₂ optical determination through flexible battery-less NFC tag" has been extensively revised and most of the questions have been addressed. Minor revisions are still needed. In particular:

- The sentence "While standard facemasks simply act as air filters for the nasal and/or mouth passage, the integration of sensors to control parameters of interest can be an added value to improve their usage and effectiveness, including the developed smart facemask into the wearable devices of the Internet of Things (IoT) paradigm." is still confusing. I would suggest to remove "including the developed smart facemask into the wearable devices of the Internet of Things (IoT) paradigm."

Response: Following the reviewer's recommendation, we have removed the suggested part of the sentence.

- A full justification of the cost estimate should be included in the SI.

Response: A new Table has been included in the SI (Supplementary Table 4), with estimations of each part/component cost according to current market prices. Many thanks for your enlightening and encouraging comments that have helped to significantly improve the manuscript.

● **Reviewer #2**

The authors made significant improvement of the manuscript based on the comments and it is ready for publication.

● **Reviewer #3**

The list of suggestions and items to improve from the prior version are all revised in the new version.

Responses: Many thanks for your enlightening and encouraging comments, which have helped to significantly improve the manuscript.